# MeInTime: Bridging Age Gap in Identity-Preserving Face Restoration

## Abstract

To better preserve an individual's identity, face restoration has evolved from reference-free to reference-based approaches, which leverage high-quality reference images of the same identity to enhance identity fidelity in the restored outputs. However, most existing methods implicitly assume that the reference and degraded input are age-aligned, limiting their effectiveness in real-world scenarios where only cross-age references are available, such as historical photo restoration. This paper proposes MeInTime, a diffusion-based face restoration method that extends reference-based restoration from same-age to cross-age settings. Given one or few reference images along with an age prompt corresponding to the degraded input, MeInTime achieves faithful restoration with both identity fidelity and age consistency. Specifically, we decouple the modeling of identity and age conditions. During training, we focus solely on effectively injecting identity features through a newly introduced attention mechanism and introduce Gated Residual Fusion modules to facilitate the integration between degraded features and identity representations. At inference, we propose Age-Aware Gradient Guidance, a training-free sampling strategy, using an age-driven direction to iteratively nudge the identity-aware denoising latent toward the desired age semantic manifold. Extensive experiments demonstrate that MeInTime outperforms existing face restoration methods in both identity preservation and age consistency. Our code is available at: `https://anonymous.4open.science/r/MeInTime-DBF7/`.

## 1 Introduction

Imagine a forensic investigation where a restored face aids in identifying a criminal, yet appears decades younger than the individual at the time. Could such a face mislead, rather than assist? Now look at a restored face in an old family album. The features say it's your beloved one, but the age doesn't match your memory—the lines deeper, the youth gone. Can such a face truly take you back in time? These scenarios, along with tasks like historical photo enhancement, cross-age verification, and long-term personal archives, underscore a critical insight: faithful face restoration requires not only identity preservation, but also age consistency.

While most existing Blind Face Restoration(BFR) methods primarily focus on identity preservation, recent advances have witnessed a shift from reference-free to reference-based paradigms. Compared to reference-free methods (Zhou et al., 2022; Wang et al., 2023; Yue & Loy, 2024; Hu et al., 2020)—which restore faces directly from various degraded inputs (*e.g.*, blur, noise, low resolution, compression artifacts)—reference-based methods (Varanka et al., 2024; Liu et al., 2025; Ying et al., 2024; Hsiao et al., 2024; Zhang et al., 2025) go further by leveraging high-quality reference images from the same identity to recover identity-faithful results even under extreme degradations, and have thus become the prevailing approach.

However, these methods are inherently limited to same-age settings, *i.e.*, the age gap between the degraded and reference image is minimal. Since the degraded input lacks reliable age cues, merely rely on reference image to compensate for missing information risks direct feature copying, potentially causing noticeable age drift when the age gap between degraded input and reference image is large, as illustrated in Fig. 1. This obvious limitation hinders the generalization of reference-based approaches to cross-age restoration in practical applications, as previously discussed.

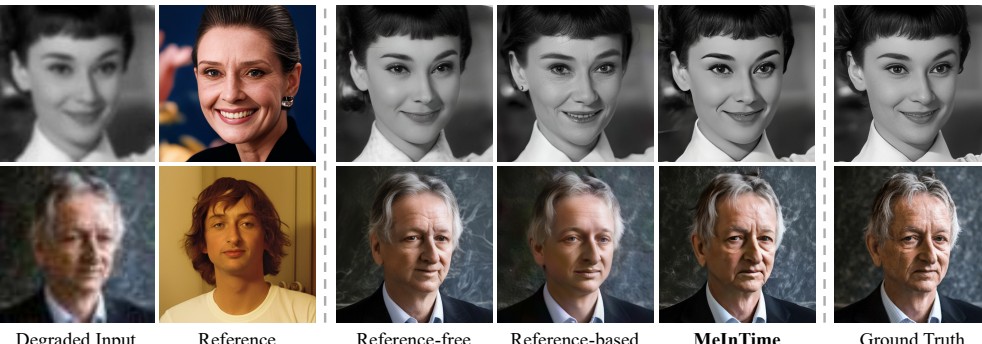

| Degraded Input | Reference | Reference-free | Reference-based | **MeInTime** | Ground Truth |

Figure 1: Given degraded inputs and cross-age references, reference-free method (Lin et al., 2024) (ECCV 2024) fails to preserve identity, while reference-based method (Liu et al., 2025) (AAAI 2025) tends to overfit reference features, causing noticeable age drift. In contrast, **MeInTime** achieves identity-faithful and age-consistent restoration. **Zoom in for best view.**

To bridge this gap, we propose MeInTime, aiming to extend reference-based face restoration from same-age to cross-age settings. Given one or few reference images along with a target age prompt with respect to the degraded input, MeInTime achieves faithful restoration with both identity fidelity and age consistency. Rather than redesigning from scratch, we use a reference-free restoration model DiffBIR (Lin et al., 2024) as our foundation. The base model is capable of producing realistic restorations, and, being built upon Stable Diffusion (Rombach et al., 2022), naturally supports text-prompted generation. Our strategy is to extend it by injecting identity features from reference images and incorporating the target age in form of text prompt as condition, enabling identity-preserving and age-consistent restoration. This process introduces three key challenges: *1) How to effectively inject identity information and target age condition? 2) How to mitigate conflicts between the implicit age cues in reference images and the desired target age? 3) How to address the scarcity of training data with age-span identity pairs?*

The proposed MeInTime employs a systematic approach to jointly address the aforementioned challenges. Our key solution is to decouple the processing of identity and age conditions across training and inference stages. Joint conditioning during training is problematic: Identity and age are inherently entangled, implicit age cues in reference images may conflict with the target age and misguide the model's learning objective, while the scarcity of cross-age identity-paired training data further limits the model's ability to learn disentangled representations. (we present a detailed dataset analysis in the Appendix B.) During training, we focus solely on identity preservation. Following (Wang et al., 2024; Papantoniou et al., 2024), we adopt a face encoder to extract enhanced identity embeddings from one or few reference images and apply them through a decoupled attention mechanism equipped with a unified text prompt "photo of a person" to guide the restoration. As the base model injects degraded images via ControlNet (Zhang et al., 2023), serving as structural guidance into the UNet decoder, potential conflicts may arise with identity features. To address this, we introduce Gated Residual Fusion modules, which dynamically learn to integrate structural features and identity cues in a content-aware manner. During inference, we propose Age-Aware Gradient Guidance, a training-free sampling strategy that steers the generative process toward the desired age manifold. Specifically, we compute an age-directional gradient in the feature space using paired prompts (*e.g.*, "photo of a person" vs. "photo of a 24-year-old person"), where the target age (*e.g.*, 24) is derived from the input age condition. And then apply it to refine the latent at each sampling step through gradient-descent algorithm. As the gradient's computation solely depends on the age attribute difference, such a guidance captures the meaningful age semantic prior while effectively canceling out identity-related information. Additionally, a timestep-scaled modulation term is introduced to adaptively control the guidance strength, facilitating restoration quality.

Our key contributions can be summarized as follows:

- We present the first reference-based face restoration framework that enables identity-preserving restoration in cross-age scenarios.
- We propose a disentangled strategy that integrates identity features through training and enhances age information through text semantics during inference, effectively mitigating identity-age conflicts and cross-age data scarcity.

- MeInTime introduces several novel designs, including robust identity embedding extraction and injection, the proposed Gated Residual Fusion module, and an plug-and-play Age-Aware Gradient Guidance technique.

- Extensive experiments demonstrate that our method outperforms existing approaches in terms of visual quality, identity preservation, and age consistency.

## 2 RELATED WORK

**Reference-free Face Restoration.** Reference-free face restoration refers to common BFR task that aims to recover high-quality face images from degraded inputs with unknown degradation. Most GAN-based methods (Wang et al., 2021; Gu et al., 2022; Zhou et al., 2022) exploit pre-trained models such as StyleGAN2 (Karras et al., 2020) or a learned VQ codebook of facial features as priors to synthesize realistic facial details (Zhao et al., 2022). Recently, diffusion-based methods (Wang et al., 2023; Yue & Loy, 2024; Yang et al., 2023; Lin et al., 2024) exploit strong generative priors of diffusion models for high-fidelity restoration. However, since these methods proceed without reference images, the restored results may deviate from the authentic identity, particularly under severe degradation.

**Reference-based Face Restoration.** Reference-based face restoration aims to enhance identity preservation by leveraging high-quality reference images of the same identity. Alignment-based methods (Li et al., 2020; 2022) extract identity-specific details by aligning reference and degraded inputs. Diffusion-based method (Varanka et al., 2024) learns personalized representations from a few reference samples but requires per-identity tuning. To avoid test-time tuning, recent approaches (Liu et al., 2025; Ying et al., 2024; Zhang et al., 2025; Hsiao et al., 2024) use face recognition models to extract identity embeddings, and integrate them via modified attention mechanism to achieve tuning-free restoration. However, these methods generally assume a minimal age gap between the reference and degraded input, limiting their effectiveness in cross-age restoration where age consistency cannot be guaranteed.

**Personalization in Diffusion Models.** Building upon advances in diffusion models, personalization methods enable the generation of specific visual concepts (*e.g.*, objects, animals) as indicated by reference images. Such conceptions are frequently adopted when the concept is a specified face identity. Some related research (Wang et al., 2024; Li et al., 2024b; Yan et al., 2023) fine-tune the model to internalize identity-specific representations from references, enabling personalized image generation. To support fine-grained control, recent work (Brack et al., 2024) injects text-driven local guidance into the denoising trajectory via DDIM-inversion, while another line of research (Li et al., 2024a) aligns StyleGAN's latent space with Stable Diffusion so that linear attribute edits can be faithfully expressed during generation. Personalized editing method (Wei et al., 2024) leverages diffusion priors to define editing-direction losses that enhance editability while preserving identity. Inspired by these advances, we extend personalized generation to the restoration domain, enabling identity-preserving and age-controllable face restoration.

## 3 METHODOLOGY

### 3.1 OVERVIEW

In this section, we outline the design of MeInTime. Our method leverages the ControlNet trained in DiffBIR (Lin et al., 2024), which encodes degraded images as structural guidance for the restoration task. The core idea is to adapt the reference-free restoration model for identity-preserving, age-consistent face restoration. Our approach can be divided into two stages. We start by training a reference-based model that focuses on identity preservation. Once applied, we leverage it in the inference stage to delve into the mechanism of age control generation. Details are discussed in the following subsections. An overview of our framework is illustrated in Fig. 2.

### 3.2 IDENTITY PRESERVATION

Due to the lack of sufficient cross-age identity datasets, it is challenging to disentangle age information from identity features in a fully supervised manner (see Appendix B for datasets analysis). In

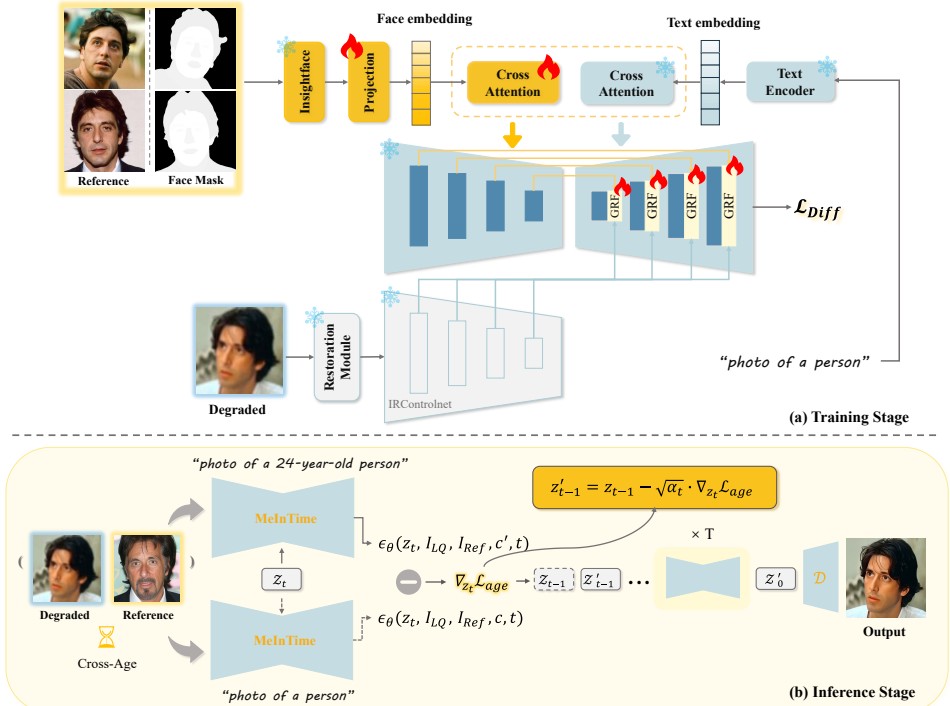

Figure 2: **Overview of MeInTime.** (a) During training, identity features from reference images are extracted by Insightface (Deng et al., 2019), projected into face embeddings, and injected into UNet via decoupled cross-attention with a unified prompt "photo of a person". Gated Residual Fusion (GRF) modules are introduced into each decoder block to facilitate feature fusion. The degraded input is processed following DiffBIR, which incorporates a restoration model (Liang et al., 2021) and the proposed IRControlNet as structural guidance. We adopt the standard diffusion loss ($\mathcal{L}_{\text{Diff}}$) as the training objective. (b) During inference, given the target age, the framework generates an age prompt (*e.g.*, "photo of a 24-year-old person") and performs two forward passes–with and without the age condition–to compute an age-aware gradient that iteratively refines the latent along the denoising process, enabling identity-preserving, age-controllable restoration.

light of this limitation, we make a deliberate compromise by first training a reference-based identity-preserving face restoration model on available identity datasets. This ensures that identity fidelity is preserved and not easily affected by the subsequent age manipulation. The architecture of this model is illustrated in Fig. 2(a).

Specifically, the model takes as input a synthesized degraded image $I_{\text{LQ}}$, $N$ reference images $I_{\text{Ref}} = \{x_i\}_{i=1}^N$ belonging to the same identity, and a unified text prompt $c =$ "photo of a person". The training objective is then given by:

$$\mathcal{L}_{\text{Diff}} = \mathbb{E}_{z_t,\, t,\, I_{\text{LQ}},\, I_{\text{Ref}},\, \epsilon}\ \|\epsilon - \epsilon_\theta(z_t, c, I_{\text{LQ}}, I_{\text{Ref}})\|_2^2, \tag{1}$$

where $z_t$ denotes the latent code at timestep $t$, $\epsilon$ the sampled noise from an Isotropic Gaussian distribution, $\epsilon_\theta(\cdot)$ the proposed model and $c$ the conditioning text embedding.

**Face Embedding.** Unlike prior works (Li et al., 2024b; Ye et al., 2023) that adopt a CLIP image encoder, we employ a face recognition (FR) model for identity feature extraction. Trained on millions of identities, the FR model captures discriminative identity cues that remain stable under age-related changes, thus reducing the influence of reference image age on extracted features. To eliminate the influence of background textures, we first apply a face parsing model (Yu et al., 2018) to extract facial region masks. Each masked face image $x_i^{\text{mask}}$ is then fed into the FR model $\phi(\cdot)$ to obtain a more neat and discriminative 512-$D$ identity embedding $e^i = \phi(x_i^{\text{mask}})$. Given a set of reference images $\{x_i\}_{i=1}^N$, we obtain the corresponding identity embeddings $\{e^i\}_{i=1}^N$, which are then averaged to form a single, robust identity representation $e = \frac{1}{N}\sum_{i=1}^N e^i$. This fused identity embedding serves as a reliable identity descriptor and is mapped into a sequence of feature tokens via a projection

network $\mathcal{P}$ to align with the embedding space of the UNet's cross-attention modules:

$$f = \mathcal{P}(e), \quad f \in \mathbb{R}^{N \times D}, \tag{2}$$

here, $N$ denotes the number of tokens and $D$ is the feature dimension. We set $N = 16$ to enable finer-grained identity conditioning.

**Feature Injection.** To effectively integrate the identity embedding $f$ into the generation process while preserving the semantic prior of the text-to-image model, we adopt a strategy similar to IP-Adapter (Ye et al., 2023) by employing dual cross-attention mechanism. Specifically, we introduce two learnable projection layers $W_k^f$ and $W_v^f$ to perform additional Attention $(Q, K^f, V^f)$, where $K^f = W_k^f \cdot f$ and $V^f = W_v^f \cdot f$. These parameters are optimized simultaneously with our projection network. Then, the output of the cross-attention is fused with the original text-based attention, allowing for the injection of identity information:

$$Out = \text{Attention}(Q, K, V) + \lambda \text{Attention}(Q, K^f, V^f), \tag{3}$$

where $\lambda$ is a balancing hyperparameter and set as 0.75 during training.

**Gated Residual Fusion.** When directly injecting identity features, we observe that training becomes unstable and the results are unsatisfactory. We hypothesize that this instability arises from conflicts between structural features from degraded input and the newly introduced identity features. In our base model, structural features $F_{\text{LQ}}^i$ are extracted by ControlNet and directly added to the skip features $F_{\text{skip}}^i$ at each decoder layer of UNet. Since identity cues have been newly incorporated into the skip features, this direct addition inevitably leads to undesired interference, as shown in Fig. 11.

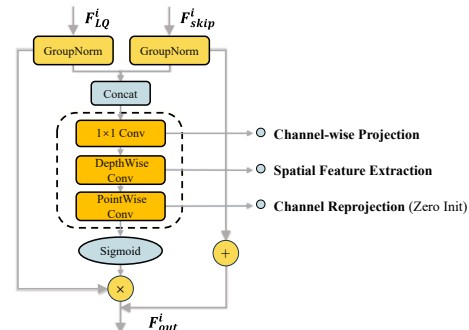

Figure 3: The structure of Gated Residual Fusion module.

To resolve this, we propose Gated Residual Fusion (GRF) module to regulate the fusion process, as shown in Fig. 3. The key insight is to learn a gating weight that adaptively modulates the influence strength of $F_{\text{LQ}}^i$ when injecting identity features. Different from standard AdaLN (Perez et al., 2018) or MLP modules, we employ Depthwise Separable Convolution (Chollet, 2017) to reduce parameters and enhance computational efficiency. The GRF module operates as follows:

$$F^i = \text{Concat}\left(\text{GroupNorm}(F_{\text{LQ}}^i), \text{GroupNorm}(F_{\text{skip}}^i)\right), \tag{4}$$

$$G^i = \text{Sigmoid}\left(\text{PWConv}\left(\text{DWConv}\left(\text{Conv}(F^i)\right)\right)\right), \tag{5}$$

$$F_{\text{out}}^i = F_{\text{skip}}^i + G^i \odot F_{\text{LQ}}^i, \tag{6}$$

where $\odot$ denotes element-wise multiplication. In this manner, the output $F_{\text{out}}^i$ serves as a fused skip connection intermediate feature for the corresponding decoder block.

### 3.3 Age-Aware Gradient Guidance

**Age Gradient Computation.** After obtaining a restoration model with strong identity preservation. Intuitively, one might leverage age prompt to query the model for age-specific generation. However, empirical observations suggest that the controllability of text prompt is somewhat weakened, as illustrated in Fig. 4. We posit that this issue stems from efforts to enhance the strength of image prompt, which may inadvertently impair the model's response to semantic signals from text prompt. To activate age priors in the text-to-image model, we first introduce age-specific

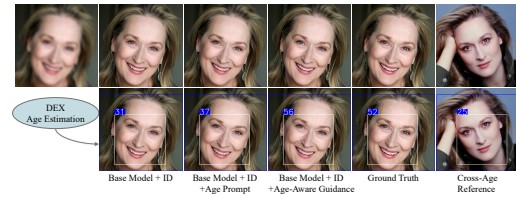

Figure 4: Visual comparison of identity-preserving restoration under different age control strategies.

---

**Algorithm 1** Age-Aware Gradient Guidance inference

---

**Input:** Degraded $I_{\text{LQ}}$; reference images $I_{\text{Ref}}$; age prompt $c'$ and generic prompt $c$ (both encoded by the CLIP text encoder); diffusion steps $T$; model $\epsilon_\theta$ (UNet integrated with GRF); VAE decoder $\mathcal{D}$
**Output:** restored image $\mathcal{D}(\tilde{z}_0)$

1:  Sample $z_T \sim \mathcal{N}(0, I)$
2:  **for** $t = T$ **to** 1 **do**
3:  $\quad z_{t-1} \leftarrow \sqrt{\alpha_{t-1}}\left( z_t - \dfrac{\sqrt{1-\alpha_t} \cdot \epsilon_\theta(z_t, I_{\text{LQ}}, I_{\text{Ref}}, c', t)}{\sqrt{\alpha_t}} \right)$   $\triangleright$ perform DDIM denoising
4:  $\quad$ **for** $n = 1$ **to** $N$ **do**
5:  $\quad\quad$ Let $z_t = z_t.\texttt{detach().requires\_grad()}$   $\triangleright$ stop-gradient on $z_t$
6:  $\quad\quad$ $\epsilon_{\text{src}} \leftarrow \epsilon_\theta(z_t, I_{\text{LQ}}, I_{\text{Ref}}, c, t)$
7:  $\quad\quad$ $\epsilon_{\text{trg}} \leftarrow \epsilon_\theta(z_t, I_{\text{LQ}}, I_{\text{Ref}}, c', t)$
8:  $\quad\quad$ $\Delta\epsilon \leftarrow \epsilon_{\text{trg}} - \epsilon_{\text{src}}$   $\triangleright$ age-specific residual
9:  $\quad\quad$ $\mathcal{L}_{\text{age}} \leftarrow (\Delta\epsilon \cdot z_t).\texttt{mean}()$   $\triangleright$ yielding explicit direction
10: $\quad\quad$ $\tilde{z}_{t-1} \leftarrow z_{t-1} - \lambda \cdot \sqrt{\alpha_t} \cdot \nabla_{z_t}\mathcal{L}_{\text{age}}$   $\triangleright$ time-scaled correction
11: $\quad$ **end for**
12: **end for**
13: **return** $\mathcal{D}(\tilde{z}_0)$
**Note:** $\lambda$ is a scaling factor to amplify gradient strength.

---

prompt in the form "photo of a $[\alpha]$-year-old person", where $\alpha$ is the input age in numerals. This follows prior work (Chen & Lathuilière, 2023a) showing that numeral-based expressions better capture age characteristics than coarse prompts (*e.g.*, "man in his thirties") or vague descriptors. Then following the score-based view of diffusion models (Song & Ermon, 2019; Song et al., 2020b), we regard the UNet as a conditional score estimator: its output gives a direction in the latent space that increases the conditional likelihood of the current latent under the given conditions. Formally,

$$\nabla_{z_t} \log p_t\big(z_t \mid I_{\text{LQ}}, I_{\text{Ref}}, c\big) \approx -\frac{1}{\sigma_t}\, \epsilon_\theta\big(z_t, I_{\text{LQ}}, I_{\text{Ref}}, c, t\big), \tag{7}$$

where $\sigma_t$ is determined by the diffusion noise schedule and $c$ is text prompt. Motivated by this view, we isolate the age attribute by contrasting the age-specific prompt $c'$ with generic prompt $c$ ("photo of a person") to provide an age-aware gradient defined as:

$$\nabla_{z_t}\mathcal{L}_{\text{age}} = \epsilon_\theta(z_t, I_{\text{LQ}},\, I_{\text{Ref}}, c', t) - \epsilon_\theta(z_t, I_{\text{LQ}},\, I_{\text{Ref}}, c, t), \tag{8}$$

This gradient residual captures the direction pointing from the model's prediction $z_t$ conditioned on $c'$ to the prediction conditioned on $c$. Following the similar spirit of prompt-to-prompt editing methods (Hertz et al., 2023; Nam et al., 2024; Wei et al., 2024), the intuition provided for such a gradient is that this residual cancels out components unrelated to the age attribute prompt—such as identity features—thus enabling high-level conceptual guidance focused purely on age semantics.

**Timestep-Scaled Latent Optimization.** Then, we leverage this gradient to refine the latent variable $z_{t-1}$ sampled from $z_t$. Considering that the sampled latent code $z_{t-1}$ at large timesteps (*e.g.*, $t = T$) contains limited facial semantic reconstruction information, the corresponding gradient estimation may become inaccurate, causing misleading guidance. Thus we introduce a timestep-scaled modulation term $\sqrt{\alpha_t}$, consistent with the

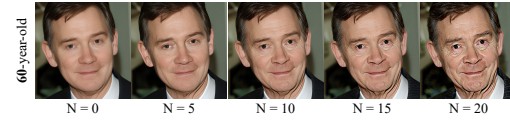

Figure 5: Visual comparison of different optimization steps.

definition in typical diffusion process, to adaptively regulate the strength of guidance across different timesteps. The calculation of the updated $\tilde{z}_{t-1}$ is as follows:

$$z_{t-1} = \sqrt{\alpha_{t-1}}\left( \frac{z_t - \sqrt{1-\alpha_t}\,\epsilon_\theta(z_t, I_{\text{LQ}},\, I_{\text{Ref}}, c', t)}{\sqrt{\alpha_t}} \right), \tag{9}$$

$$\tilde{z}_{t-1} = z_{t-1} - \sqrt{\alpha_t} \cdot \nabla_{z_t}\mathcal{L}_{\text{age}}, \tag{10}$$

we update $z_{t-1}$ at every denoising step following the DDIM sampling trajectory (Song et al., 2020a). Regarding the selection of optimization steps, the target attribute–human age–is a subtle and fine-grained concept, where overly aggressive optimization can easily distort identity or introduce artifacts. Through extensive experiments, we find that setting $N = 5$ consistently yields the best age

guidance performance, achieving a good balance between controllability and visual fidelity, as illustrated in Fig. 5. By incrementally nudging $z_{t-1}$ along the direction of the gradient, the generated image exhibits greater fidelity to the desired age prompt while maintaining identity consistency. The whole algorithm is illustrated in Algorithm 1.

## 4 EXPERIMENTS

### 4.1 EXPERIMENTAL SETUP

**Training Datasets.** Training is conducted on VGGFace2-HQ (Cao et al., 2018) and CelebRef-HQ (Li et al., 2022), both containing identity-consistent images captured within narrow age variation. A total of 5,405 identities are selected, yielding 178,877 images. All images are centrally aligned and resized to a spatial resolution of $512^2$. For each training instance, we randomly choose 1∼5 images with the same identity as reference images. We adopt a widely used first-order degradation pipeline to generate the corresponding low-quality counterparts. See Appendix D for details.

**Testing Datasets.** We evaluate our method on two testing sets representing same-age and cross-age scenarios. For **same-age** testing, we select 150 identities from the remaining individuals in CelebRef-HQ, using the same setup as training and setting "photo of a person" as global prompt to focus evaluation on identity fidelity. For **cross-age** testing, we adopt the AgeDB dataset (Moschoglou et al., 2017), which comprises 16,488 facial images of 568 celebrities, each annotated with integer age labels ranging from 0 to 101 using the age estimator DEX (Rothe et al., 2015). Given that all images are low-resolution ($112^2$), we apply a super-resolution method (Chen et al., 2020) to upscale them to $512^2$. And identity consistency is ensured by filtering samples with ArcFace model (Deng et al., 2019). Based on this procedure, we construct a curated subset of 100 identities. The average age gap between the ground-truth and reference images is 26 years, and the ground-truth age label is used to generate age prompt for cross-age guidance.

**Implementation Details.** We use Stable Diffusion 2.1-base integrated with DiffBIR (Lin et al., 2024) components as our base restoration model. The model is finetuned for 250K iterations using the AdamW (Loshchilov & Hutter, 2017) optimizer with a learning rate of 4e-5. Training is conducted on three NVIDIA RTX 4090 GPUs with a batch size of 8 per GPU. To enable classifier-free guidance (Ho & Salimans, 2022), we randomly drop identity embeddings with a probability of 0.05. The guidance scale is set to 1 during training and increased to 4 during inference.

### 4.2 COMPARISONS WITH STATE-OF-THE-ART METHODS

**Compared Methods.** We compare MeInTime with state-of-the-art baselines. For reference-free restoration, we evaluate CodeFormer (Zhou et al., 2022), DR2+SPAR (Wang et al., 2023) and DifFace (Yue & Loy, 2024). In terms of reference-based approaches, we include all publicly available methods with released code, namely DMDNet (Li et al., 2022), Ref-LDM (Hsiao et al., 2024), RestorerID (Ying et al., 2024), and FaceMe (Liu et al., 2025). As RestorerID supports only a single reference, we select the first image from each identity's reference set for a fair comparison.

**Evaluation Metrics.** The analysis is conducted from three perspectives: image quality, identity preservation, and age consistency. Image quality is measured using PSNR, SSIM, and LPIPS (Zhang et al., 2018) (full-reference), as well as FID (Heusel et al., 2017), and MUSIQ (Ke et al., 2021) (no-reference). Identity similarity (IDS) is measured by cosine similarity of ArcFace (Deng et al., 2019) embeddings. Age consistency is quantified by predicting restored ages with age estimator DEX Rothe et al. (2015) and calculating mean absolute error (MAE) against ground-truth labels, denoted as AGE.

**Evaluation on Same-Age Data.** As shown in Tab. 1(left), MeInTime achieves the best performance on PSNR, LPIPS, and IDS, and ranks second on SSIM and MUSIQ, demonstrating superior identity preservation along with enhanced pixel and perceptual quality. Qualitative results in Fig. 6 further validate these findings. In the first row, despite severe degradation, our method effectively leverages reference identity cues while maintaining the facial structure of the degraded. In the second row, even with a slightly non-frontal input, MeInTime produces faithful result, including precise reconstruction of fine details such as accessories.

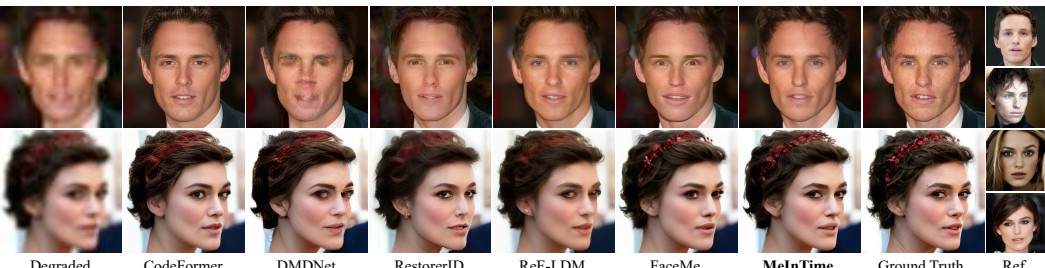

Degraded | CodeFormer | DMDNet | RestorerID | ReF-LDM | FaceMe | **MeInTime** | Ground Truth | Ref.

Figure 6: Comparison with state-of-the-art methods on same-age data. **Zoom in for best view.**

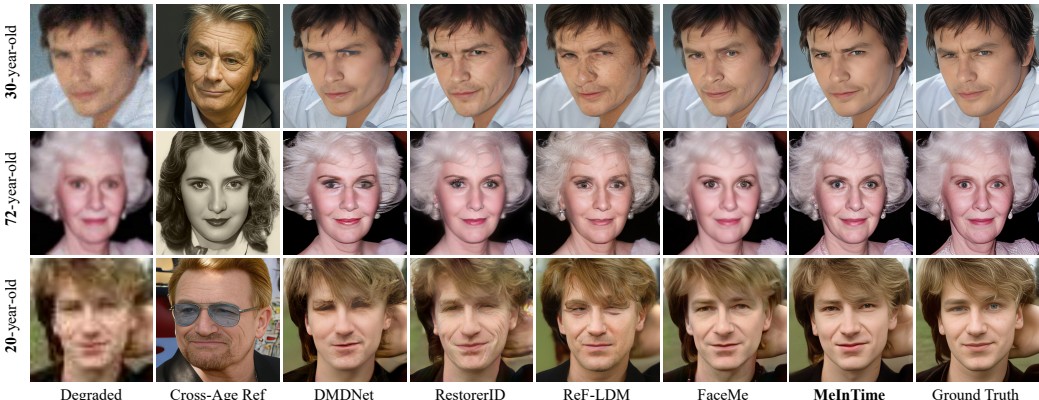

Degraded | Cross-Age Ref | DMDNet | RestorerID | ReF-LDM | FaceMe | **MeInTime** | Ground Truth

Figure 7: Comparison with state-of-the-art methods on cross-age data. **Zoom in for best view.**

**Evaluation on Cross-Age Data.** Tab. 1(right) presents quantitative results on cross-age dataset with significant age gaps. MeInTime achieves top performance in MUSIQ, IDS, and AGE, and ranks second in LPIPS and FID. Notably, it significantly surpasses all baselines in age accuracy, demonstrating superior age alignment. Qualitative comparisons in Fig. 7 clearly reveal that existing reference-based methods suffer from age drift, while our approach effectively corrects age deviations. Even under the compounded challenge of both large age gap and facial occlusion (row 3), MeInTime maintains robust performance, highlighting its resilience and generalization. Fig. 8 shows MeInTime excels in both identity preservation and age consistency.

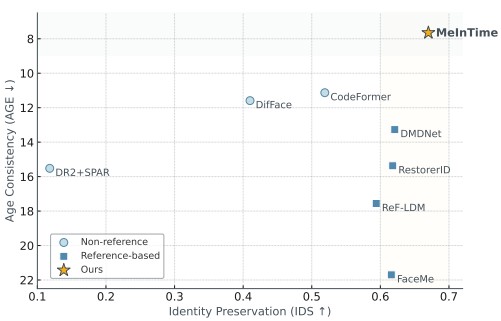

Figure 8: Identity Preservation vs. Age Consistency.

**User Study.** We consider face restoration a human-centric task, particularly because our goal aims to distinguish fine facial attributes such as identity and age. Objective metrics alone are insufficient to capture how users actually perceive these nuances, so we conducted a user study to obtain a more reliable human-centered assessment. We selected 20 cross-age restoration pairs and compared our method against four competitive baselines. 50 volunteers participated in the study. The questionnaire covered three dimensions: visual quality, identity similarity, and age consistency. For each question, participants selected the result that best satisfied the specified criterion. The complete questionnaire is provided in Appendix K. The results are summarized in Fig. 9. Although MeInTime scores only slightly lower than CodeFormer in visual quality (28.8% vs. 31.2%), it substantially outperforms all baselines in identity similarity (37.8%, the highest among all methods) and achieves a significant lead in age consistency, receiving 64.5% of all votes, which is +45.0 percentage points higher than the second-best method. These results indicate that MeInTime maintains high visual quality while delivering the most faithful identity preservation and the most reliable age control in cross-age restoration.

Table 1: Quantitative comparison with state-of-the-art methods on same-age and cross-age face restoration. The best results are shown in yellow, and the second-best are blue.

| Method | Ref | Same-Age | | | | | | Cross-Age | | | | | | |
|---|---|---|---|---|---|---|---|---|---|---|---|---|---|---|
| | | PSNR↑ | SSIM↑ | LPIPS↓ | MUSIQ↑ | FID↓ | IDS↑ | PSNR↑ | SSIM↑ | LPIPS↓ | MUSIQ↑ | FID↓ | IDS↑ | AGE↓ |
| CodeFormer | | 25.61 | 0.720 | 0.207 | 75.87 | 47.46 | 0.639 | 26.29 | 0.749 | 0.142 | 75.77 | 55.85 | 0.519 | 11.13 |
| DR2+SPAR | | 21.97 | 0.681 | 0.312 | 71.31 | 69.43 | 0.213 | 21.71 | 0.646 | 0.357 | 64.52 | 95.05 | 0.118 | 15.52 |
| DifFace | | 25.31 | 0.720 | 0.247 | 70.00 | 55.01 | 0.509 | 24.94 | 0.691 | 0.274 | 65.31 | 83.73 | 0.410 | 11.59 |
| DMDNet | ✓ | 25.76 | 0.730 | 0.228 | 73.92 | 55.48 | 0.703 | 25.76 | 0.712 | 0.233 | 72.94 | 66.37 | 0.621 | 13.27 |
| RestorerID | ✓ | 25.00 | 0.699 | 0.272 | 72.75 | 59.74 | 0.686 | 24.94 | 0.679 | 0.242 | 75.71 | 69.18 | 0.618 | 15.37 |
| ReF-LDM | ✓ | 24.80 | 0.713 | 0.227 | 73.55 | 46.46 | 0.714 | 24.58 | 0.677 | 0.236 | 75.55 | 63.58 | 0.594 | 17.56 |
| FaceMe | ✓ | 26.41 | 0.733 | 0.220 | 73.01 | 47.34 | 0.722 | 26.23 | 0.722 | 0.219 | 74.68 | 60.70 | 0.616 | 21.70 |
| **MeInTime** | ✓ | 26.50 | 0.731 | 0.200 | 75.51 | 47.99 | 0.733 | 24.75 | 0.719 | 0.211 | 76.66 | 60.01 | 0.670 | 7.65 |

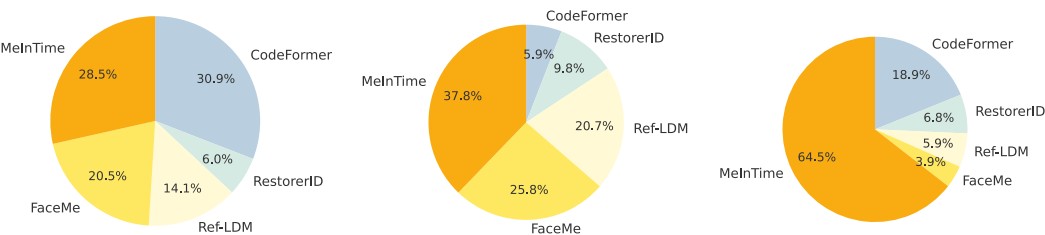

(a) Visual quality preference.        (b) Identity similarity preference.        (c) Age consistency preference.

Figure 9: User study results.

## 4.3 ABLATION STUDIES

**Robustness on Varying Age Gaps.** To guarantee valid references across all age intervals, we select 30 identities from the cross-age testing dataset. For each identity, reference images are divided into five age-gap intervals relative to the degraded input, yielding five restorations per identity across increasing age disparities. As shown in Tab. 2, all metrics remain stable across all intervals, demonstrating that MeInTime is resilient to large age gaps without compromising visual quality, identity, or age consistency.

**Inference Strategy.** Tab. 3 presents the comparison of proposed Age-Aware Gradient Guidance inference strategy with direct age prompt conditioning. While both strategies yield similar perceptual quality and identity preservation, the age prompt alone fails to enforce accurate age control (AGE 14.30). In contrast, our gradient-based guidance achieves a significantly lower AGE score (7.65), demonstrating more effective age-specific restoration. Visual comparison is shown in Fig. 10.

**Effectiveness of Gated Residual Fusion.** We assess the impact of proposed GRF module by comparing models trained with and without it. Since GRF focuses on identity-preserving, we report the results on same-age testing. As shown in Tab. 4, the absence of GRF leads to degraded performance across all metrics, particularly a drop in IDS. Qualitative results in Fig. 11 reveal that without GRF, the restored output lacks critical identity details, such as missing eyeglasses and oversmoothed hair textures, whereas GRF enables more visually natural and identity-preserving restoration.

**Timestep-Scaled Modulation Term.** We evaluate the proposed timestep-scaled modulation in Age-Aware Gradient Guidance by comparing fixed scales (0, 0.5, 1.0) against our adaptive strategy $\sqrt{\alpha_t}$. As shown in Fig. 12, fixed scale of 0 or 0.5 yield insufficient guidance, causing residual feature copying from reference, while a scale of 1.0 introduces overcorrection and artifacts. In contrast, the adaptive $\sqrt{\alpha_t}$ modulates the guidance strength according to the diffusion sampling timestep, striking a balance between semantic precision and visual fidelity for better restoration.

## 5 CONCLUSION

This paper presents MeInTime, the first reference-based face restoration framework extending identity-preserving restoration from same-age to cross-age settings. It supports arbitrary numbers of

Table 2: Ablation study on varying age gaps.

| Age-Gap | PSNR↑ | SSIM↑ | LPIPS↓ | IDS↑ | AGE↓ |
|---|---|---|---|---|---|
| ≤ 10 | 23.92 | 0.711 | 0.213 | 0.676 | 4.19 |
| 10∼20 | 23.87 | 0.710 | 0.210 | 0.680 | 4.30 |
| 20∼30 | 23.97 | 0.710 | 0.215 | 0.674 | 5.47 |
| 30∼40 | 24.36 | 0.715 | 0.210 | 0.681 | 4.51 |
| >40 | 23.83 | 0.708 | 0.211 | 0.679 | 4.72 |

Table 3: Ablation study on different inference methods.

| Inference Strategy | PSNR↑ | SSIM↑ | LPIPS↓ | IDS↑ | AGE↓ |
|---|---|---|---|---|---|
| Age Prompt | **25.37** | 0.712 | 0.211 | 0.667 | 14.30 |
| Age Guidance | 24.75 | **0.719** | 0.211 | **0.670** | **7.65** |

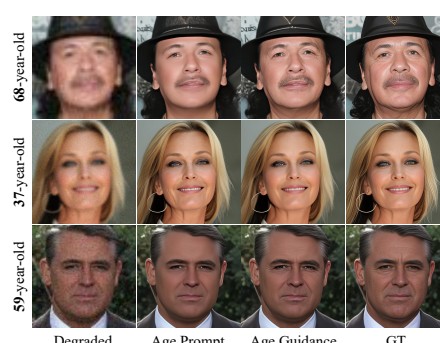

Figure 10: Visual comparison of different inference methods.

Table 4: Ablation study on GRF module.

| GRF Module | PSNR↑ | SSIM↑ | LPIPS↓ | IDS↑ |
|---|---|---|---|---|
| w/o GRF | 25.12 | 0.713 | 0.237 | 0.670 |
| w/ GRF | **26.50** | **0.731** | **0.200** | **0.733** |

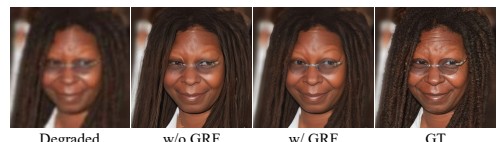

Figure 11: Visual comparison with and without GRF module.

Figure 12: Visual comparison with different guidance scales.

reference images, enabling identity-preserving restoration while generating age-specific semantics conditioned on the target age. Extensive experiments show that MeInTime achieves state-of-the-art performance in visual quality, identity fidelity, and age consistency, offering a promising solution for faithful face restoration. Our limitations and future improvements are discussed in the Appendix I.

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

APPENDIX

# A PRELIMINARIES

## A.1 DIFFUSION MODELS

Diffusion models (Ho et al., 2020) learn a data distribution by reversing a fixed Gaussian noising process. Let $\{\beta_t\}_{t=1}^{T}$ be the variance schedule with $\alpha_t = 1 - \beta_t$ and $\bar{\alpha}_t = \prod_{i=1}^{t} \alpha_i$. The forward (diffusion) transition at step $t$ is

$$q(z_t \mid z_{t-1}) = \mathcal{N}\big(\sqrt{\alpha_t}\, z_{t-1},\, (1 - \alpha_t)\, \mathbf{I}\big), \tag{11}$$

which yields the reparameterization

$$z_t = \sqrt{\alpha_t}\, z_{t-1} + \sqrt{1 - \alpha_t}\, \epsilon, \qquad \epsilon \sim \mathcal{N}(0, \mathbf{I}), \tag{12}$$

and the closed form $z_t = \sqrt{\bar{\alpha}_t}\, z_0 + \sqrt{1 - \bar{\alpha}_t}\, \epsilon$.

During sampling, we start from $z_T \sim \mathcal{N}(0, \mathbf{I})$ and iteratively denoise. An $\epsilon$-parameterized denoiser $\epsilon_\theta(\cdot)$ predicts the noise component for the noisy latent $z_t$ under condition $\mathcal{C}$. We adopt the deterministic DDIM (Song et al., 2020a) update to compute the previous latent from $z_t$:

$$z_{t-1} = \sqrt{\alpha_{t-1}}\left( \frac{z_t - \sqrt{1 - \alpha_t}\, \epsilon_\theta(z_t, \mathcal{C}, t)}{\sqrt{\alpha_t}} \right), \tag{13}$$

Repeating denoising trajectory produces $z_0$, which is finally decoded to the image domain by the VAE decoder used in the main model.

## A.2 SCORE-BASED VIEW

A convenient way to view diffusion models is through score matching (Song et al., 2020b): instead of modeling a density directly, we learn its score—the gradient of the log-density with respect to the noisy latent.

Let $p_t(z_t \mid \mathcal{C})$ denote the conditional distribution of the latent at step $t$ under condition $\mathcal{C}$. The UNet output is proportional to the conditional score:

$$\nabla_{z_t} \log p_t(z_t \mid \mathcal{C}) \approx -\frac{1}{\sigma_t} \epsilon_\theta(z_t, \mathcal{C}, t). \tag{14}$$

Thus, $\epsilon_\theta(\cdot)$ provides a direction in the latent space that increases the likelihood under $\mathcal{C}$.

## A.3 SCORE DISTILLATION SAMPLING

Score Distillation Sampling (SDS) (Poole et al., 2022) is a general technique that leverages a pretrained diffusion model as a guidance signal to optimize external variables. Instead of training the denoiser itself, SDS uses the score field learned by the diffusion model to provide gradients that guide another generative process or directly optimize the latent variable.

Formally, given an input latent $z$, condition $y$, a pretrained denoiser $\epsilon_\theta$, a randomly sampled timestep $t$, and noise $\epsilon \sim \mathcal{N}(0, \mathbf{I})$, the diffusion loss is defined as:

$$\mathcal{L}_{\text{Diff}}(z, y, \epsilon, t) = \big\| \epsilon_\theta(z_t, y, t) - \epsilon \big\|_2^2. \tag{15}$$

For a variable $z$ that depends on parameters $\phi$ (e.g., $z = g_\phi(\cdot)$, where $g$ is a generator), the chain rule gives

$$\nabla_\phi \mathcal{L}_{\text{Diff}} = \Big( \epsilon_\theta(z_t, y, t) - \epsilon \Big) \frac{\partial \epsilon_\theta(z_t, y, t)}{\partial z_t} \frac{\partial z_t}{\partial \phi}. \tag{16}$$

Prior work shows that omitting the UNet Jacobian term (the middle factor) yields an effective gradient for distillation, leading to the SDS gradient

$$\nabla_\phi \mathcal{L}_{\text{SDS}}(z, y, \epsilon, t) = \Big( \epsilon_\theta(z_t, y, t) - \epsilon \Big) \frac{\partial z_t}{\partial \phi}. \tag{17}$$

In image editing tasks (Hertz et al., 2023; Nam et al., 2024), one often optimizes the latent directly (no external generator), *i.e.*,

$$\nabla_{z_t} \mathcal{L}_{\text{SDS}} \;=\; \epsilon_\theta(z_t, y, t) - \epsilon. \tag{18}$$

In this sense, our proposed age-guided sampling method can also be viewed as a variant of SDS.

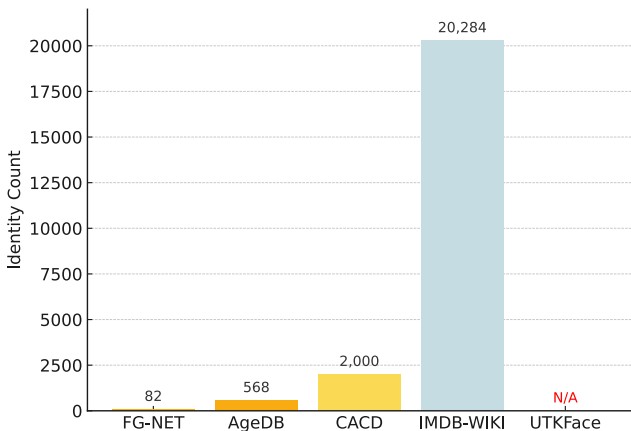

(a) Number of identities in each dataset.

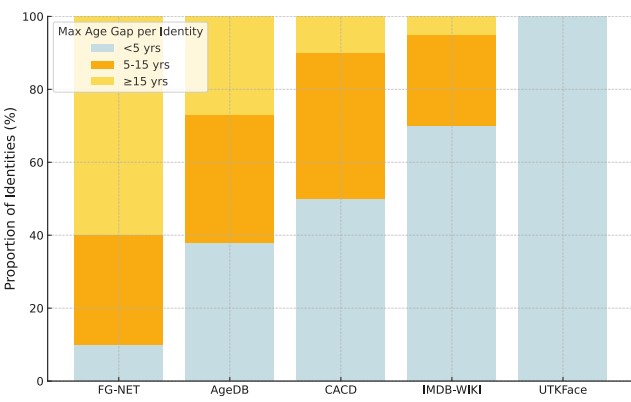

(b) Identity age span distribution in datasets.

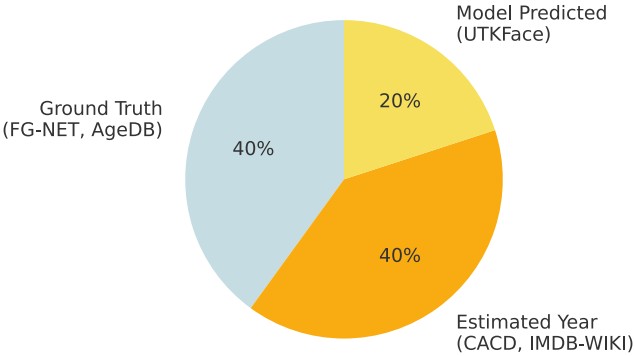

(c) Age label type distribution across datasets.

Figure 13: Statistics of cross-age identity datasets.

## B    ANALYSIS OF CROSS-AGE IDENTITY DATASETS

In this section, we conduct a focused analysis of five representative cross-age identity datasets (FG-NET (Panis et al., 2016), AgeDB (Moschoglou et al., 2017), CACD (Chen et al., 2014), IMDB-WIKI (Rothe et al., 2018), UTKFace (Zhang et al., 2017)) to reveal key limitations that hinder joint modeling of identity and age conditions. Based on the observations as visualized in Fig. 13, we summarize three core challenges, as discussed below.

**Insufficient Number of Identities** Fig. 13a shows the total number of labeled identities in each dataset. FG-NET and AgeDB contain only 82 and 568 identities respectively, making them inadequate for training models that require identity diversity. CACD improves this with 2,000 identities, while IMDB-WIKI contains a large number (over 20,000), but many are noisy or weakly verified. UTKFace lacks identity annotations entirely, and thus cannot be used for identity-paired training.

**Limited Age Span per Identity** Fig. 13b presents the distribution of maximum age gap within each identity. These proportions are estimated based on available statistics and prior literature. Notably, a majority of identities in FG-NET and AgeDB have age spans under 15 years, and identities with $\geq 15$ years gap–critical for cross-age modeling–remain rare across all datasets. This scarcity limits the model's ability to observe meaningful aging patterns within the same individual.

**Unreliable Age Annotations** Fig. 13c illustrates the proportion of samples with different types of age labels. Only FG-NET and AgeDB provide ground-truth annotations (40%). CACD and IMDB-WIKI rely on estimated birth years (40%), while UTKFace employs predicted ages from age estimation models (20%). The lack of consistently accurate labels introduces ambiguity during training and impairs age-controlled generation.

These observations demonstrate that existing datasets are insufficient to fully support cross-age identity-paired modeling due to limited identity diversity, constrained intra-identity age variation, and unreliable labels. To address these limitations, our method separates identity and age supervision across training and inference. During inference, we adopt AgeDB as our testing dataset, as it offers relatively clean identity labels, large span age gaps, and verified age annotations, making it a suitable benchmark for age-guided generation.

## C    DETAILS ABOUT AGE-AWARE GRADIENT GUIDANCE OPTIMIZATION

**Mitigating Artifacts via Prompt Adjustment** We observe that directly applying optimization guidance during restoration can lead to artifacts similar to those encountered in Score Distillation Sampling methods, such as over-sharpening, unnatural textures, and distorted facial details. To alleviate this, we adopt a strategy inspired by (McAllister et al., 2024), which mitigates undesired artifacts by appending textual descriptors of typical artifacts to the source prompt. Following this idea, we enrich the generic prompt with mild negative descriptors (*e.g.*, "oversaturated," "blurry," "cartoon-like," "malformed") to implicitly discourage such effects during guidance. As shown in Eq. 19, by subtracting enriched prompt $c$, the guidance signal $\Delta\epsilon$ implicitly penalizes these artifact-prone features and steers the restoration process away from them. This prompt-level contrast enables lightweight but effective suppression of visual degradation, without requiring additional training or model modification.

$$\Delta\epsilon = \epsilon_\theta(z_t, I_{\text{LQ}}, I_{\text{Ref}}, c', t) - \epsilon_\theta(z_t, I_{\text{LQ}}, I_{\text{Ref}}, c, t), \tag{19}$$

## D    EXPERIMENT IMPLEMENTATION DETAILS

**Degraded Image Synthesis** To simulate real-world image degradation during training, following (Lin et al., 2024), we adopt the widely used first-order degradation pipeline that simulates real-world distortions through a combination of blur, downsampling, upsampling, noise injection, and compression. Concretely, given a high-quality image $I_{hq}$, we synthesize the degraded version $I_{lq}$ as follows:

$$I_{lq} = \left\{ \left[ (I_{hq} \otimes k_\sigma) \downarrow_r + n_\delta \right]_{\text{JPEG}_q} \right\} \uparrow_r, \tag{20}$$

where $\otimes$ denotes convolution, $k_\sigma$ denotes a Gaussian kernel with standard deviation $\sigma$, $\downarrow_r$ and $\uparrow_r$ represent bicubic downsampling and upsampling operations with scale factor $r$, $n_\delta$ is Gaussian noise

with standard deviation $\delta$, and $[\cdot]_{\text{JPEG}_q}$ denotes JPEG compression with quality $q$. The parameters $\sigma$, $r$, $\delta$, and $q$ are independently sampled from $[0.2, 10]$, $[1, 12]$, $[0, 15]$, $[30, 100]$, respectively.

**Design of testing data instance** For the same-age test set, we set it up in the same way as building the training data pairs, that is, one degraded image (LQ), randomly select 1-5 images with the same identity as reference images. For the cross-age test set, we deliberately enforce an age gap between the LQ image and its references (see Fig. 14): for each LQ image, we sample 1–5 same-identity reference images whose ages differ from target age substantially. Across 100 identities, the resulting mean absolute age gap between the target and references is 26 years.

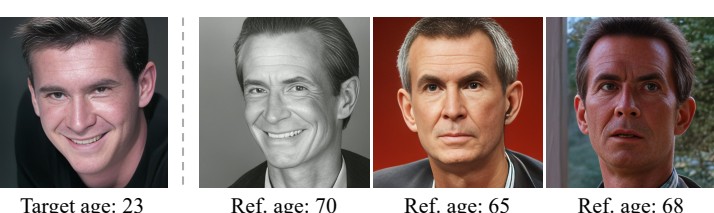

| Target age: 23 | Ref. age: 70 | Ref. age: 65 | Ref. age: 68 |

Figure 14: Cross-age testing data instance.

# E    MORE QUALITATIVE COMPARISONS

To supplement the qualitative analysis in the main paper, we provide more comparison results with baseline methods on both same-age and cross-age datasets. As shown in Fig. 19, Fig. 20, our method consistently produces more visually realistic restorations with better alignment to the desired identity and age, demonstrating robust generalization across varying reference conditions.

# F    STABILITY OF IDENTITY CONDITIONING UNDER AGE-AWARE GRADIENT GUIDANCE

To further verify that the identity embeddings learned during training remain stable and unaffected when performing age-aware gradient guidance at inference, we conduct an additional ablation study focusing on the robustness of the identity condition when reference images cover different age gaps.

For this analysis, we fix a degraded input image and its target age prompt, and select reference images of the same identity spanning different age range relative to the target age. We construct 20 such sets in total. For each set, every reference image is used in an independent run with age-aware gradient guidance, while the degraded input and target age remain unchanged. During each run, we visualize the cross-attention map corresponding to the identity tokens over all sampling steps and average them to obtain a single heatmap, which reveals how identity cues modulate the latent throughout inference.

Fig. 15 shows one representative visualization of the result. The identity-token attention maps exhibit nearly identical spatial patterns, consistently focusing on identity-relevant facial regions such as the eyes, nose, and overall facial structure. This stability indicates that the injected identity embeddings are largely invariant to the age of the reference images and remain unaffected by the age-aware guidance. It also aligns with the design of Eq. 8 that the age gradient cancels out components unrelated to the age attribute prompt such as identity features. The observation is consistent with the quantitative ablation in Tab. 2, where identity similarity remains stable across increasing age-gap groups. Together, these findings demonstrate that identity and age are effectively disentangled: identity features are preserved, while the age-guided updates influence only the intended age-related semantics.

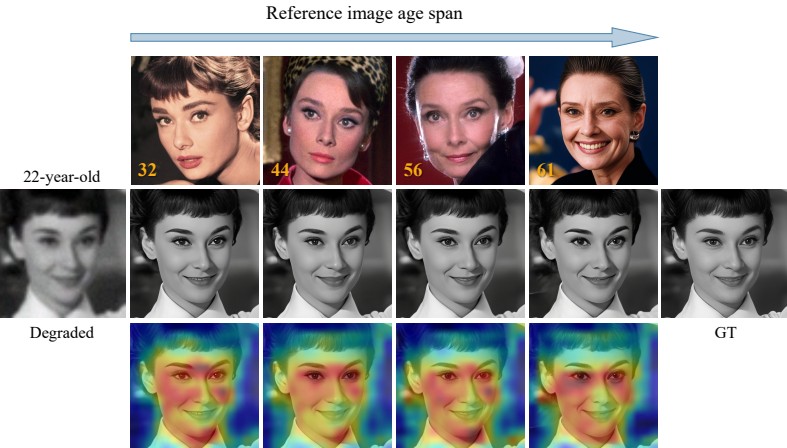

Figure 15: **Visualization of identity-token attention maps.** Each column shows one restoration run with a different-age reference of the same identity. The second row gives the restored results, and the bottom row shows the averaged 16×16 identity-token attention maps across all steps.

## G   EVALUATION ON THE REAL-WORLD DATASET

To complement the synthetic experiments and more faithfully reflect real-world conditions, we further evaluate our method on real-world cross-age images.

Since no existing benchmark provides in-the-wild identity datasets with reliable age annotations, we construct a dedicated real-world evaluation set. Specifically, we collect low-quality face images of 20 public figures from online sources, including low-resolution media images or video frames. The approximate shooting age of each image is inferred from publicly available information and time-stamped records. For each identity, we then gather high-quality images taken at least 20 years apart to ensure that the reference images capture age-discriminative facial features.

For evaluation metrics, as ground-truth images are unavailable, image quality is evaluated by null-reference metrics, namely MUSIQ and FID. Identity similarity (IDS) is computed between restored images and the reference images. And AGE metric is estimated by calculating the mean absolute error between DEX-predicted age of the restored image and the provided target age.

Quantitative results are reported in Tab. 5 MeInTime achieves the best identity preservation among all methods, reaching an IDS of 0.477, and also obtains the lowest AGE score (7.43), demonstrating superior alignment with the target age. In terms of perceptual quality, MeInTime delivers competitive MUSIQ performance while achieving the best FID score (54.06). Representative qualitative comparisons are shown in Fig. 16, where MeInTime produces sharper structures, more faithful identity traits, and more natural age characteristics, validating its effectiveness under real-world degradation conditions.

Table 5: Quantitative comparison on the real-world dataset. The best results are shown in yellow, and the second-best are blue.

| Method | Ref | MUSIQ ↑ | FID ↓ | IDS ↑ | AGE ↓ |
|---|---|---|---|---|---|
| CodeFormer | | 70.17 | 56.53 | 0.263 | 9.67 |
| DR2+SPAR | | 65.09 | 123.50 | 0.229 | 10.23 |
| DifFace | | 56.28 | 119.78 | 0.295 | 11.83 |
| DMDNet | ✓ | 69.79 | 105.54 | 0.295 | 9.67 |
| RestorerID | ✓ | 74.25 | 76.31 | 0.467 | 15.14 |
| Ref-LDM | ✓ | 68.04 | 59.93 | 0.352 | 13.33 |
| FaceMe | ✓ | 65.00 | 80.99 | 0.372 | 9.00 |
| **MeInTime** | ✓ | 74.18 | 54.06 | 0.477 | 7.43 |

Figure 16: Comparison with state-of-the-art methods on real-world data. **Zoom in for best view.**

## H  EVALUATION ON TWO-STAGE RESTORATION–EDITING PIPELINES

To examine whether cross-age restoration can be achieved through a two-stage pipeline that first restores the face and then performs age editing on the restored result, we conduct an additional experiment. Specifically, we randomly select 15 identities from the cross-age testing set and first obtain their restored images using reference-based methods RestorerID Ying et al. (2024), Ref-LDM Hsiao et al. (2024), and FaceMe Liu et al. (2025). We then apply three widely used age-editing models—HRFAE Yao et al. (2021), SAM Alaluf et al. (2021), and FADING Chen & Lathuilière (2023b)—to these restored images, conditioning each method on the corresponding target age.

As illustrated in Fig. 17, the post-editing results reveal clear limitations of this two-stage pipeline. HRFAE often produces mild and incomplete age changes that fail to reach the target age. SAM, in contrast, frequently applies overly aggressive transformations that distort facial structure or introduce visible artifacts, sometimes failing entirely. FADING generates noticeable age shifts but tends to drift away from the reference identity or produce unrealistic facial attributes. These observations indicate that post-hoc age editing is inherently prone to error accumulation: once identity cues from the reference have been fused into the restored image, subsequent editing easily disrupts this balance, either by weakening identity fidelity or by introducing age features in a structurally inconsistent manner.

Complementing the qualitative findings, the quantitative results (editing on FaceMe outputs) in Tab. 6 further validate that all three age-editing methods exhibit clear drops in visual quality and identity similarity when applied to restored images. Compared with the only diffusion-based editor Fading, MeInTime achieves more efficient in runtime (43.37s vs. 133.41s). These results highlight the advantages of our unified framework, which eliminates the cascading errors observed in two-stage pipelines and enables reliable cross-age restoration without compromising visual quality or identity fidelity.

Table 6: Quantitative assessment of post-editing effects on FaceMe-restored images.

| Method | Type | PSNR ↑ | LPIPS ↓ | MUSIQ ↑ | IDS ↑ | AGE ↓ | Time(s) |
|--------|------|--------|---------|---------|-------|-------|---------|
| HRFAE | GAN | 21.57 | 0.308 | 67.31 | 0.602 | 10.23 | 0.18 |
| SAM | GAN | 18.09 | 0.457 | 60.33 | 0.347 | 8.77 | 0.45 |
| FADING | Diffusion | 22.28 | 0.270 | 66.52 | 0.584 | 8.05 | 133.41 |
| **MeInTime** | **Diffusion** | **24.53** | **0.208** | **74.33** | **0.655** | **6.89** | **43.37** |

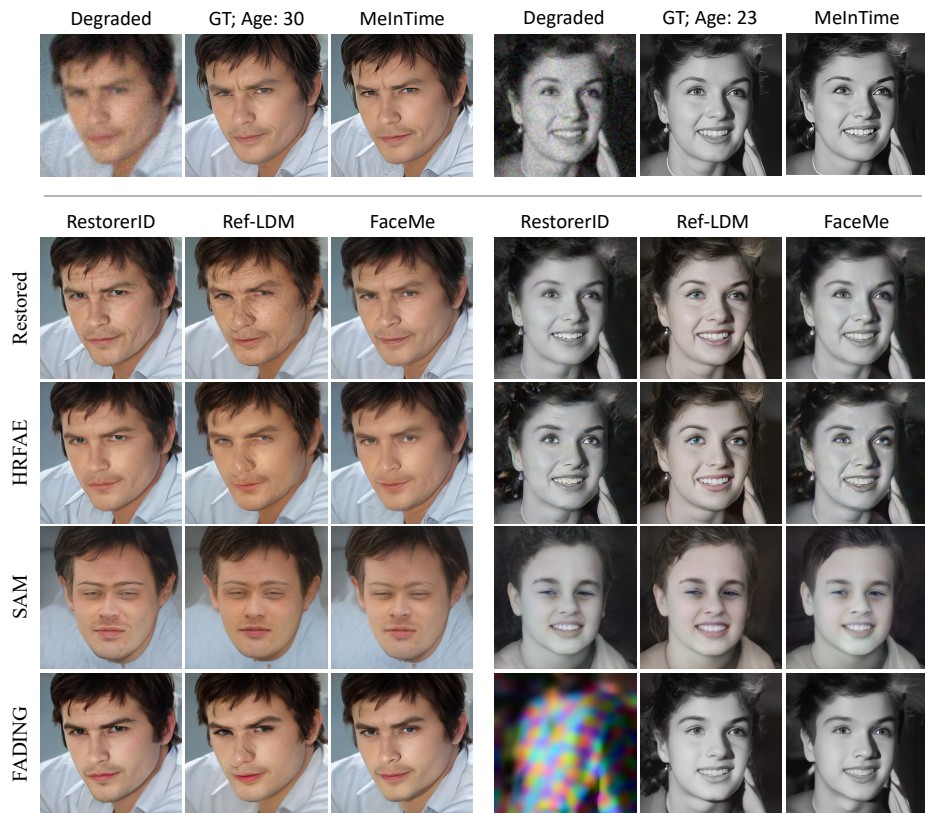

Figure 17: Post-hoc age editing applied to restored images from RestorerID, Ref-LDM, and FaceMe. Existing age-editing models either under-edit, over-edit, or damage identity, demonstrating the limitation of two-stage pipelines for cross-age restoration.

# I  LIMITATIONS

Despite the promising results, our approach still has several limitations. First, when multiple reference images of the same identity span a wide range of ages, the aggregated identity embedding may be biased, leading to suboptimal restoration fidelity. Second, the age control mechanism relies on textual prompts to encode age semantics; however, such prompts may not always align with the actual visual perception of age, and when conditioning on very high age targets, the model may occasionally produce over-sharpened or exaggerated facial textures, as illustrated in Fig. 18. Third, our Age-Aware Gradient Guidance introduces extra optimization iterations during inference, increasing sampling time compared to standard single-pass approaches. Tab. 7 reports the inference time of all diffusion-based methods using 50-step DDIM sampling, where $N = 0$ corresponds to same-age restoration and $N = 5$ denotes the number of cross-age optimization steps in MeInTime. Inspired by the one-step diffusion model (Sauer et al., 2024; Yin et al., 2024; Wang et al., 2025), we will consider training the distillation diffusion model to achieve a single time step optimization. In future work, we aim to enhance the robustness of identity representation under age variation, explore more efficient guidance mechanisms, and investigate continuous age control with finer granularity.

Table 7: Inference time comparison.

| Methods | DifFace | DR2 | RestorerID | Ref-LDM | FaceMe | MeInTime (N=0) | MeInTime (N=5) |
|---|---|---|---|---|---|---|---|
| Time(s) | 5.96 | 1.23 | 7.74 | 1.79 | 7.65 | 7.26 | 43.37 |

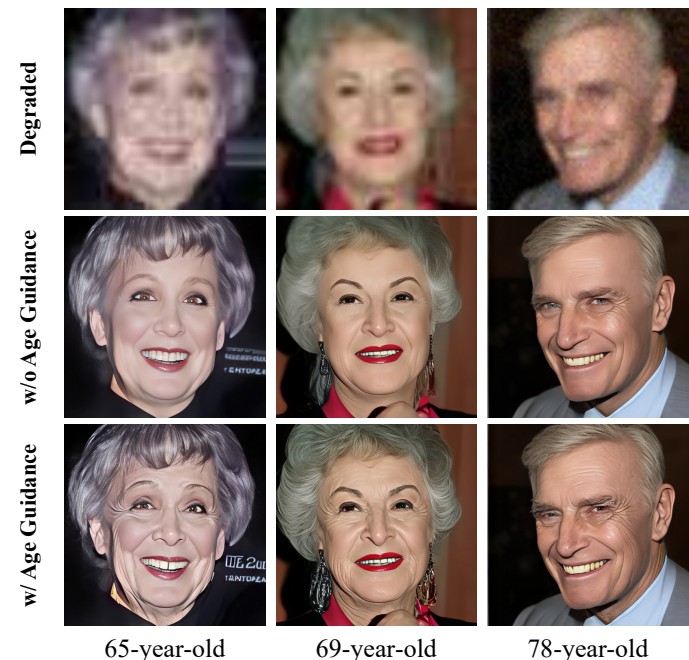

Figure 18: Over-sharpening artifacts when conditioning on high-age prompts.

## J LLM USAGE

Large Language Models (LLMs) were employed solely to assist with the linguistic refinement of this manuscript. In particular, we used an LLM to improve readability and clarity by performing tasks such as sentence rephrasing, grammar checking, and polishing the overall flow of the text.

The LLM was not involved in formulating research questions, developing methodology, conducting experiments, or analyzing data. All scientific ideas, technical contributions, and results presented in this paper were entirely conceived and verified by the authors. The use of the LLM was limited to enhancing the presentation quality of the manuscript.

The authors retain full responsibility for all aspects of the paper. Any text suggested or polished by the LLM was manually reviewed and revised to ensure accuracy and compliance with ethical standards. The LLM is not considered an author, and its use does not contribute to plagiarism or scientific misconduct.

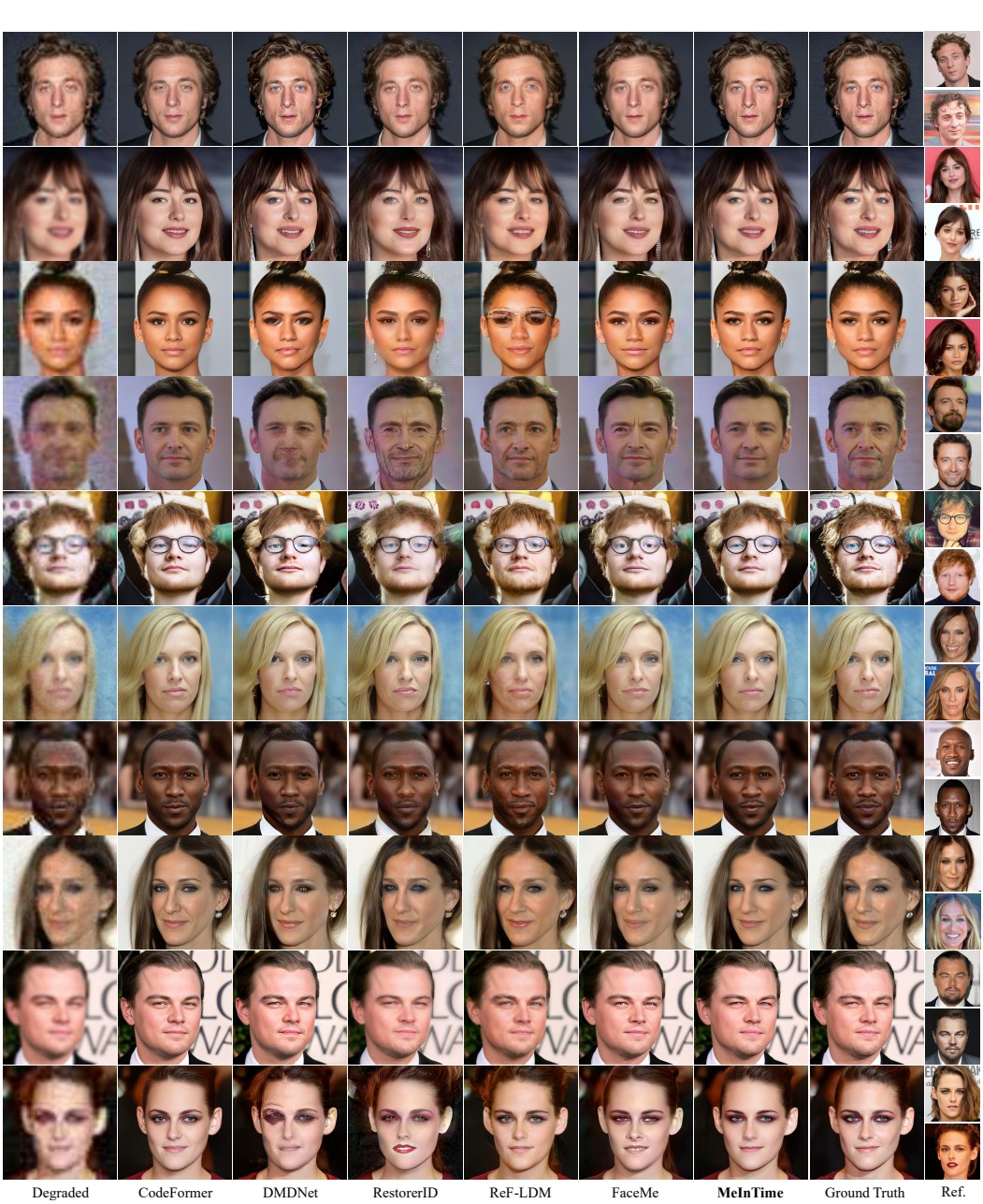

Figure 19: Comparison with state-of-the-art methods on same-age data. **Zoom in for best view.**

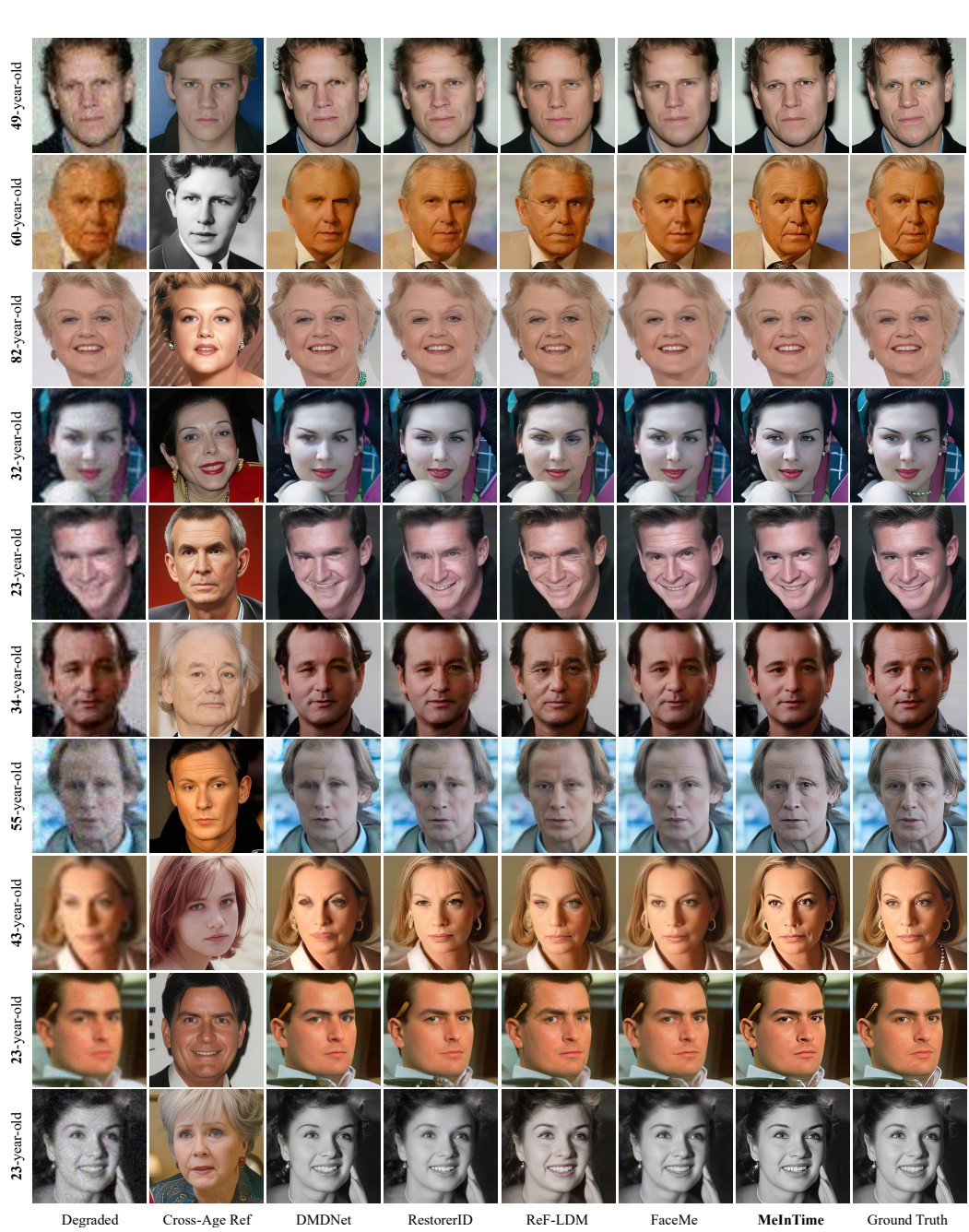

Figure 20: Comparison with state-of-the-art methods on cross-age data. **Zoom in for best view.**

## K  COMPLETE USER STUDY CONTENT

This questionnaire presents several face restoration results generated by different methods for the same input. For each question, please select the result that best matches the described criterion. Each question focuses on a single evaluation dimension; please follow the provided guidance carefully.

I. Visual Quality

**Q1. Which result has the best overall visual quality?**

Consider: image sharpness, presence of artifacts, distortion, and overall naturalness.

II. Identity Similarity

**Q2. Which result most resembles the same person as the ground-truth (GT) image?**

Focus: facial structure, shape, consistent facial features (*e.g.*, eyes, pupils, eyebrows, nose contour), and recognizable identity traits.

**Q3. Which result most resembles the identity in the reference image?**

Focus: whether the identity remains recognizable despite age differences.

III. Age Consistency

**Q4. Which result best matches the age appearance of the GT image?**

Consider: skin texture, wrinkles, smoothness, and aging marks consistent with the GT image.

**Q5. Which result best matches the target age described in the age prompt?**

Consider: whether the generated age-related features accurately correspond to the target age specified in the prompt.

Target Age: 30-year-old

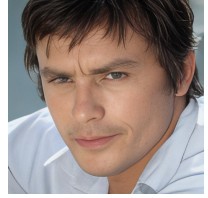 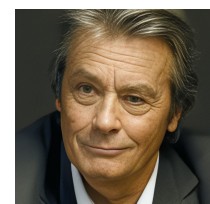

**Ground Truth**  **Reference**

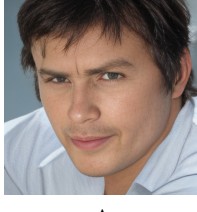 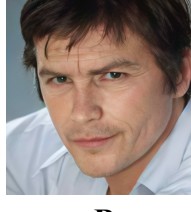 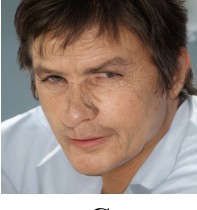 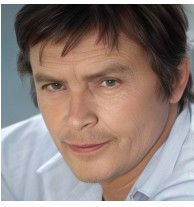 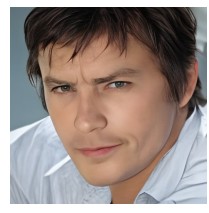

**A**  **B**  **C**  **D**  **E**

Figure 21: **Example set used in the user study.** For each question, participants are shown the ground-truth image, a reference image, and multiple restored results (A–E). They select the option that best satisfies the evaluation criterion for that question. The target age indicated at the top (*e.g.*, "30-year-old") is used only for age-related questions.

