# OpenReview forum: "MeInTime: Bridging Age Gap in Identity-Preserving Face Restoration"
_ICLR.cc/2026/Conference — Submitted to ICLR 2026_

### Official Review · Reviewer_4pTD · 2025-10-27

**Soundness:** 3
**Presentation:** 3
**Contribution:** 2
**Rating:** 6
**Confidence:** 4

**Summary:**

This paper presents MeInTime, a diffusion-based face restoration framework designed to handle cross-age reference-based restoration. Traditional reference-based methods assume age alignment between degraded and reference images, which limits their applicability in real-world scenarios (e.g., historical photos). MeInTime decouples the modeling of identity and age: during training, identity features are learned through reference embeddings and Gated Residual Fusion modules; during inference, an Age-Aware Gradient Guidance strategy is introduced to steer restoration toward the desired age manifold without retraining. Experiments on same-age and cross-age datasets show improved identity preservation and age consistency compared to existing baselines.

**Strengths:**

+ The paper extends reference-based face restoration to the cross-age domain, which is an underexplored but practically relevant scenario.

+ Separating identity learning and age guidance is conceptually elegant and avoids conflicts between identity and age signals.

+ The Gated Residual Fusion module is well-motivated and effectively stabilizes identity–structure fusion.

+ The authors benchmark on both same-age and cross-age datasets and introduce age-consistency metrics, providing clear evidence of quantitative gains.

+ Visual examples demonstrate that the model generates more age-consistent restorations compared to prior works.

**Weaknesses:**

– Most degraded samples used for visualization and evaluation are synthetically generated with severe distortions (e.g., heavy blur or compression). Under such extreme degradation, facial wrinkles and texture cues are largely lost, making it unreliable to infer age semantics from low-quality inputs. This could easily cause instability or misalignment between estimated and target ages in practice.

– The method introduces a gradient-based optimization process during inference to achieve age control. However, it is not clear whether this method is better than those using face editing-based solutions on the reference image or the restored images. It remains unclear why such simpler or more direct methods are not considered or compared.

– The AGE metric depends on a pretrained estimator, which might not correlate well with perceptual aging. Additional perceptual studies or human evaluations would strengthen the claims.

- The so-called ID-preserving sampling in Eq. 8 effectively modifies the denoising trajectory based on facial feature gradients, which is conceptually similar to facial attribute editing or latent direction control seen in previous works such as [r1,r2] or other latent manipulation methods. The distinction between this “sampling” and conventional attribute-based editing is not articulated. Maybe it is better to add some face editing works in Section 2.

[r1] When StyleGAN Meets Stable Diffusion: a w+ Adapter for Personalized Image Generation
[r2] LEDITS++: Limitless Image Editing using Text-to-Image Models

**Questions:**

- Many degraded samples in the paper are heavily distorted. How does the model perform under moderate degradations, where age cues are still partially available?


- Why not apply a face editing approach to either (a) edit the reference image to match the target age before restoration, or (b) perform age editing after restoration? Would this achieve comparable or even better age consistency with less computational overhead?

- How stable is the Age-Aware Gradient Guidance when the reference and degraded ages differ drastically (e.g., 20s vs. 80s)? Does it sometimes produce over-aging or artifacts?

---

> ### Author Response · Authors · 2025-11-27
> **Response to Reviewer 4pTD [Part 1/2]**
>
> We thank the reviewer for the thoughtful and constructive feedback. We have uploaded a revised version of our paper with added analyses and kindly invite the reviewer to check. Below we address each concern point by point.
>
> **Q1: Under extreme degradation where age cues in the low-quality input are unreliable, can this lead to instability or misalignment in MeInTime’s age-controlled generation?**
>
> The reviewer's concern arises from a misunderstanding of how age semantics are delivered in MeInTime. Our age-controlled generation is entirely driven by the text prompt (the input target age) rather than relying on the low-quality (LQ) image to infer age information. In our design, the age guidance during inference can be viewed as a prompt-driven editing scheme, where the age semantics come from SD’s semantic prior. The LQ image is encoded via ControlNet to provide structural guidance only, so the injected feature $F^i_{LQ}$ does not contained fine-grained semantic attributes such as age. The age semantics determined by the text and the structure information from LQ are independent in their functional purposes.
>
> In fact, under severe degradations MeInTime often has a clear advantage over other reference-based methods, where LQ features usually directly interact with reference features (e.g., RestorerID). When LQ is insufficient to provide effective information, such methods tend to rely heavily on the reference features, which can cause age drift in cross-age restoration. In contrast, our model has robustly learned to restore structural information from LQ image and incorporate identity information during training. At inference, we correct the age cues only along the age semantic direction as defined in Eq. (8): $\nabla_{z_t} L_{age} = \epsilon_\theta(z_t, I_{LQ}, I_{Ref}, c', t) - \epsilon_\theta(z_t, I_{LQ}, I_{Ref}, c, t)$
> , which deliberately avoids perturbing identity and structure (details in Sec. 3.3). This inference-time editing strategy greatly alleviates instability in age-controlled generation.
>
> **Q2: How does the model perform under moderate degradations, where age cues are still partially available?**
>
> We appreciate the reviewer’s thoughtful question. The degradations in our test set follow exactly the same degradation pipeline used during training (Appendix D “Degraded Image Synthesis”), which covers a wide range of mild, moderate, and heavy distortions. The quantitative results in Tab. 1 reflect performance across this full spectrum. The qualitative examples in the main paper mainly illustrate severely degraded cases, primarily to highlight the difficulty of cross-age restoration under challenging conditions. For a more complete view, Appendix E (Fig. 19 and Fig. 20) includes more examples, covering mild and moderate degradations, and we kindly invite the reviewer to refer to these results.
>
> In addition, we have now included experiments on real-world degraded images in Appendix G, with both quantitative and qualitative analysis. These real-world samples typically exhibit light-to-moderate degradations, and MeInTime consistently demonstrates strong performance in terms of visual quality, identity preservation, and age consistency.
>
> We thank the reviewer again for this helpful suggestion. In future work, we plan to further categorize the test data by degradation severity to provide a more fine-grained evaluation.
>
> **Q3: The AGE metric relies on a pretrained age estimator, which may not align well with human perception. Do the authors provide additional human evaluations to support their claims?**
>
> We thank the reviewer for raising this valuable point regarding the need for perceptual validation beyond estimator-based metrics. In the revised version, we have included a dedicated user study in Sec. 4.2 (“User Study”), complemented by the full questionnaire design provided in Appendix K. These subjective evaluations offer human-centered evidence for visual quality, identity similarity, and age consistency. We kindly invite the reviewer to refer to these sections for the details.

---

> > ### Author Response · Authors · 2025-11-27
> > **Response to Reviewer 4pTD [Part 2/2]**
> >
> > **Q4: Why not apply a face editing approach to either (a) edit the reference image to match the target age before restoration, or (b) perform age editing after restoration? Would this achieve comparable or even better age consistency with less computational overhead?**
> >
> > We thank the reviewer for this insightful suggestion. We conduct an additional experiment in Appendix H using a two-stage pipeline: we first restore images with reference-based methods (RestorerID, Ref-LDM, FaceMe), and then apply three widely used age-editing methods (HRFAE [Yao *et al.*, 2021], SAM [Alaluf *et al.*, 2021], FADING [Chen & Lathuilière, 2023] to performing age editing on the restored results. Details about specific experimental settings, data selection, evaluation metrics, and quantitative and qualitative analysis of experimental results please refer to the newly uploaded paper.
> >
> > To help the reviewer quickly grasp the core findings of this experiment, we present below the quantitative results of applying three age-editing methods (HRFAE, SAM, FADING) to the FaceMe-restored images:
> >
> > | Method       | Type          | PSNR ↑    | LPIPS ↓   | MUSIQ ↑   | IDS ↑     | AGE ↓    | Time(s)   |
> > | ------------ | ------------- | --------- | --------- | --------- | --------- | -------- | --------- |
> > | HRFAE        | GAN           | 21.57     | 0.308     | 67.31     | 0.602     | 10.23    | 0.18      |
> > | SAM          | GAN           | 18.09     | 0.457     | 60.33     | 0.347     | 8.77     | 0.45      |
> > | FADING       | Diffusion     | 22.28     | 0.270     | 66.52     | 0.584     | 8.05     | 133.41    |
> > | **MeInTime** | **Diffusion** | **24.53** | **0.208** | **74.33** | **0.655** | **6.89** | **43.37** |
> >
> > Overall, MeInTime outperforms all post-editing baselines by a large margin across visual quality, identity similarity, and age alignment, while also being substantially more efficient than the only diffusion-based editor (Fading). These significant differences mainly stem from the inherent capability limits of age-editing methods and the error accumulation introduced by the two-stage pipeline. This phenomenon can be clearly observed in the qualitative comparisons, and we kindly invite the reviewer to refer to Appendix H for details.
> >
> > **Q5: How stable is the Age-Aware Gradient Guidance when the reference and degraded ages differ drastically (e.g., 20s vs. 80s)? Does it sometimes produce over-aging or artifacts?**
> >
> > Our paper already includes a dedicated analysis in Sec. 4.3 “Robustness on Varying Age Gaps”, where we evaluate robustness by grouping reference images of each identity into five age-gap intervals (≤10, 10–20, 20–30, 30–40, >40 years). Tab. 2 clearly reveals that all metrics remain stable across intervals, indicating that MeInTime is resilient to varying age gaps.
> >
> > To further address the reviewer’s concern about extreme cases (e.g., 20s vs. 80s), we additionally select 8 identities from the testing set that satisfy reference–degraded pairs with 50–60-year gaps, and report the extended results below:
> >
> > | Age-Gap   | PSNR ↑    | SSIM ↑    | LPIPS ↓   | IDS ↑     | AGE ↓    |
> > | --------- | --------- | --------- | --------- | --------- | -------- |
> > | ≤10       | 23.92     | 0.711     | 0.213     | 0.676     | 4.19     |
> > | 10~20     | 23.87     | 0.710     | 0.210     | 0.680     | 4.30     |
> > | 20~30     | 23.97     | 0.710     | 0.215     | 0.674     | 5.47     |
> > | 30~40     | 24.36     | 0.715     | 0.210     | 0.681     | 4.51     |
> > | >40       | 23.83     | 0.708     | 0.211     | 0.679     | 4.72     |
> > | **50~60** | **23.77** | **0.710** | **0.212** | **0.676** | **5.17** |
> >
> > These results are still consistent with all other intervals, confirming that the Age-Aware Gradient Guidance remains stable even under extreme age differences.
> >
> > The age guidance in MeInTime is an iterative and fully controllable optimization process, as described in Sec. 3.3 “Timestep-Scaled Latent Optimization’’ and Algorithm 1. The strength of age editing can be governed by the number of optimization steps *N*. Increasing *N* moves the result closer to the age semantics but may introduce over-sharpening or artifacts as shown in Fig. 5. Through extensive experiments, we find *N* = 5 provides a stable balance across all age ranges. In practical use, if any over-aging or insufficient aging is observed, it can be easily corrected by decreasing or increasing *N*, respectively, since *N* is user-adjustable.
> >
> > **Q6: Should related work on facial editing be added to Section 2 to clarify the distinction?**
> >
> > We thank the reviewer for the helpful suggestion. We have added the relevant facial editing works (r1, r2) to Sec. 2 in the revised paper to clarify the distinction.
> >
> > **Reference:**
> >
> > - Yao *et al.*, 2021. *High Resolution Face Age Editing.*
> >
> > - Alaluf *et al.*, 2021. *Only a Matter of Style: Age Transformation Using a Style-Based Regression Model.*
> >
> > - Chen & Lathuilière, 2023. *Face Aging via Diffusion-Based Editing.*

---

> > > ### Comment · Reviewer_4pTD · 2025-11-27
> > > **Response to the authors**
> > >
> > > Thanks to the authors for such a detailed response and the revised manuscript, which has helped clarify my earlier misunderstanding of the work. I think the idea is interesting and the overall design is reasonable. Before I consider raising my score, however, I would prefer that the authors first address the concerns raised by the other reviewers.

---

> ### Author Response · Authors · 2025-11-27
> **Response to Reviewer 4pTD**
>
> Thank you very much for your warm and encouraging follow-up comment. We truly appreciate your recognition of the revised manuscript and the clarifications we provided, and we thank you again for your earlier insightful suggestions, which have been highly helpful in improving the work.
>
> We also apologize for the slight delay in responding to the reviewers’ comments. **At this point**, we have addressed all concerns raised by other reviewers and have uploaded a fully updated version of the paper accordingly.
>
> We sincerely appreciate your continued engagement and look forward to any further feedback. Thank you for your time and constructive guidance.

---

### Official Review · Reviewer_ojSk · 2025-10-29

**Soundness:** 2
**Presentation:** 3
**Contribution:** 3
**Rating:** 4
**Confidence:** 4

**Summary:**

This paper proposes MeInTime, a diffusion-based reference-guided face restoration framework aimed at handling cross-age scenarios where the reference and degraded images of the same person have large age gaps. The key idea is to decouple identity and age conditioning: identity embeddings are injected during training through a Gated Residual Fusion (GRF) module, while age consistency is adjusted during inference using a training-free Age-Aware Gradient Guidance.

**Strengths:**

Solid engineering design, the combination of GRF for stable identity fusion and age-aware gradient guidance is well implemented and empirically effective.

Clear structure and writing,the paper is clearly written, visually engaging, and the methodology section is easy to follow.

Training-free age control: The gradient-based age guidance is conceptually clean and avoids extra finetuning.

**Weaknesses:**

1. The cross-age reference setting is a rare and contrived use case. It’s unclear how many real restoration tasks actually require explicit age matching. The work does not convincingly show that this setting matters beyond a few illustrative examples. The paper starts from the observation that current reference-based face restoration methods assume the reference and degraded faces are of similar age.
While this is technically true, the practical importance of bridging “cross-age” gaps in face restoration is quite limited.

2. Experiments rely on synthetically degraded data; no evaluation on truly degraded or historical photos, which weakens the claim of “real-world generalization.”

3. The paper does not report inference time or computational overhead. Given that the proposed Age-Aware Gradient Guidance involves multiple iterative optimization steps, efficiency could be a major concern for practical applications. Quantitative timing or FLOPs comparison is missing.

4. While automatic metrics (FID, MUSIQ, AGE MAE) are presented, subjective evaluations (user or identity verification studies) are missing, which are important for human-centric tasks like face restoration.

5.The related work section overlooks several important prior approaches:

[1].Face Super-Resolution Guided by 3D Facial Priors

[2].Rethinking Deep Face Restoration

**Questions:**

1. Interestingly, Table 1 shows that on cross-age restoration, the reference-free baseline CodeFormer achieves higher PSNR and SSIM than the proposed MeInTime, despite lacking reference information. This suggests that the inclusion of reference images may actually harm reconstruction fidelity when the reference and degraded faces differ significantly in age. The authors should analyze this phenomenon more carefully, as it weakens the central claim that MeInTime “bridges” the age gap effectively.

---

> ### Author Response · Authors · 2025-11-26
> **Response to Reviewer ojSk [Part 1/2]**
>
> We thank the reviewer for the valuable comments. In the revised version, we have added (1) experiments on real-world testing set, (2) user study evaluation, and (3) runtime report. We kindly invite the reviewer to check the updated paper. Below we provide point-by-point responses to the reviewer’s concerns. Due to the character limit of the response box, we provide our replies in two parts.
>
> **Part.1**
>
> **Q1: Is the cross-age face restoration setting a reasonable and necessary problem? Is MeInTime really limited in real restoration tasks?**
>
> We will answer this question from three aspects. **(1)** While we agree that cross-age restoration occurs less frequently than same-age restoration, it represents a high-value and high-impact category of real tasks. A representative example is historical and archival footage: the degraded material was captured decades earlier, while available high-quality images of the same person come from much later in life, making the age gap unavoidable (we have added such examples to the real-world testing dataset). Similar temporal gaps arise in forensic investigations, where blurred surveillance of a long-term fugitive must be matched to an outdated ID photo, and in related scenarios such as missing-person tracing. **(2)** The age–identity entanglement we address is not an isolated assumption but a long-standing research focus. Prior work on cross-age face recognition and identity-preserving age editing (Liu *et al.*, 2025, Mi *et al.*, 2025) has consistently shown that age variation strongly affects identity fidelity. Our work brings this well-established perspective into the reference-based restoration setting, where age discrepancies have been largely overlooked. **(3)** Although MeInTime targets cross-age scenarios, it also achieves superior performance in standard same-age restoration (Sec. 4.2, "Evaluation on Same-Age Data"). Since our age-aware gradient guidance is plug-and-play, disabling it yields conventional reference-based restoration, meaning that we extend the capability to cross-age cases without sacrificing effectiveness in typical settings.
>
> **Q2: Why does the reference-free baseline CodeFormer achieve higher PSNR/SSIM than MeInTime under cross-age settings, and does this imply that reference images harm reconstruction fidelity when ages differ significantly?**
>
> We thank the reviewer for the insightful question. It is important to note that CodeFormer and MeInTime target different objectives. CodeFormer, as a reference-free early approach, focuses primarily on low-level image quality. In contrast, MeInTime is designed for a more challenging multi-objective setting—restoring visual quality while simultaneously preserving identity and achieving age consistency. Consequently, pixel-level metrics such as PSNR and SSIM tend to favor CodeFormer, which does not explicitly incorporate other constraints and therefore stays closer to the original structure in a purely structural sense. Taking all objectives into account, MeInTime substantially improves IDS (0.670 vs. 0.519) and AGE score (7.65 vs. 11.13), and compared to reference based methods with similar purposes, it also shows advantages in visual quality metrics.
>
> We analyze the slight PSNR/SSIM drop in the cross-age case arises from our age-aware gradient guidance: the iterative global update that corrects the target-age semantics inevitably spreads gradient over the whole image and may introduce mild deviations in image structures, as in Eq.(10): $z_{t-1}' = z_{t-1} - \sqrt{\alpha_t} \cdot \nabla_{z_t} L_{age}$ , which will be a focus of our future refinement.
>
> In addition, beyond pixel-level measures, perceptual quality remains central to evaluating image quality. In our newly added real-world experiments (Appendix G), MeInTime surpasses CodeFormer on perceptual measures MUSIQ and FID, suggesting that our method is more robust under real degraded conditions.
>
> **Q3: What is the performance of MeInTime in real-world scenarios?**
>
> To address this concern, we have added a comprehensive real-world evaluation in Appendix G, including dataset construction, evaluation protocol, and both quantitative and qualitative comparisons, and we kindly invite the reviewer to check the updated paper. The experiment results show that MeInTime achieves strong visual quality, identity preservation, and age consistency, demonstrating its robustness and effectiveness under real-world conditions.
>
> **Reference:**
> - Liu *et al.*, 2025. *From Cradle to Cane: A Two-Pass Framework for High-Fidelity Lifespan Face Aging.*
>
> - Mi *et al.*, 2025. *TimeMachine: Fine-Grained Facial Age Editing with Identity Preservation.*

---

> ### Author Response · Authors · 2025-11-26
> **Response to Reviewer ojSk [Part 2/2]**
>
> **Part.2**
>
> **Q4: What is the inference time?**
>
> We report the inference time of all diffusion-based restoration methods under unified 50-step DDIM sampling setup:
>
> | Methods | DifFace | Dr2  | RestorerID | Ref-LDM | FaceMe | MeInTime(N=0) | MeInTime(N=5) |
> | ------- | ------- | ---- | ---------- | ------- | ------ | ------------- | ------------- |
> | Time(s) | 5.96    | 1.23 | 7.74       | 1.79    | 7.65   | 7.26          | 43.37         |
>
> In the same-age setting (N=0), MeInTime takes 7.26s per image, which is comparable to FaceMe (7.65s) and RestorerID (7.74s). In the cross-age setting with our recommended N=5 age-guidance steps, the runtime increases to 43.37s. We think this additional cost is acceptable in practice, as cross-age restoration is generally a low-frequency but high-value scenario: typically arising in historical archive recovery or forensic identity verification, where the images are usually processed individually rather than in bulk. In such settings, achieving accurate restoration is far more critical than real-time latency.
>
> To put this overhead into perspective, it is helpful to consider the complexity of the task we tackle. Our method performs cross-age face restoration, which requires recovering faithful identity while simultaneously handling age-related variations. In this sense, the age editing procedure can be viewed as a sub-task of our approach. For reference, Fading—the most effective age editing pipeline based on DDIM inversion—requires approximately 133.41s per image (details provided in Appendix H). By comparison, our unified MeInTime framework completes the entire restoration in 43.37s, despite addressing a significantly more complex task, and is still substantially more efficient while achieving superior results.
>
> **Q5: How does the paper address the need for subjective evaluations?**
>
> We thank the reviewer for highlighting the importance of subjective evaluations. In response, we have added a dedicated user study in Sec. 4.2 ("User Study") of the revised manuscript and provided detailed questionnaire design in Appendix K. We kindly invite the reviewer to refer to these sections.
>
> **Q6: Did the original submission miss some relevant prior work?**
>
> We thank the reviewer for pointing out the missing related work. We have added the suggested references—*Face Super-Resolution Guided by 3D Facial Priors* and *Rethinking Deep Face Restoration*—to Sec. 1 and Sec. 2 in the revised paper.

---

### Official Review · Reviewer_4omw · 2025-10-31

**Soundness:** 3
**Presentation:** 3
**Contribution:** 3
**Rating:** 4
**Confidence:** 3

**Summary:**

The paper proposes MeInTime, a diffusion-based framework for identity-preserving face restoration that explicitly addresses the challenge of large age gaps between degraded input images and available reference images. MeInTime separates identity and age conditioning: it injects robust identity features during training, while, during inference, it leverages a novel Age-Aware Gradient Guidance mechanism based on textual prompts to control aging, decoupling identity from age in the generative process.

**Strengths:**

1.The paper targets a practically relevant and previously underexplored problem: high-fidelity, identity-preserving face restoration when only cross-age references are available.
2. The use of a decoupled training-inference strategy—training for identity preservation and introducing age controllability at inference—is a thoughtful response to the lack of large-scale cross-age paired datasets, as discussed with supporting data in Appendix B/Figure 12 (Page 15–16).

**Weaknesses:**

1. Limited Theoretical Analysis of Attribute Decoupling: The paper claims that identity and age are decoupled by design (identity during training, age only via gradient guidance at inference), yet it lacks a more principled investigation or proof of whether and to what extent this decoupling is reliably achieved. For example, no formal analysis or visualization is provided to demonstrate that the injected identity embeddings or the age gradients are indeed orthogonal in the learned space. Without explicit investigation, it remains somewhat speculative whether the method fully avoids entanglement between age and identity, especially under distribution shifts.

2. Structural and Optimization Details May Hinder Reproducibility: While implementation details are provided, several critical elements are only broadly sketched. For example, the precise effect of the GRF module hyperparameters, the initialization process for identity/token projection, and the inferred scaling of guidance during inference are not dissected in depth. Additionally, Algorithm 1 might benefit from being more explicit on the stopping criteria, initialization of age/generic prompts, and GRF interaction during inference.

**Questions:**

1. Can the authors provide a principled analysis (e.g., mutual information, attention visualization, or orthogonality in feature space) demonstrating that their identity embeddings remain robust (and do not leak age information) when reference images cover very wide age gaps? Empirical or theoretical clarification here would strengthen the claim of identity-age decoupling.

2. What is the actual computational overhead (in wall-clock time or FLOPs) for MeInTime during inference compared to, say, FaceMe or RestorerID, particularly under the Gradient Guidance with multiple optimization passes? Is the method practical for real-time or high-throughput use cases?

---

> ### Author Response · Authors · 2025-11-25
> **Response to Reviewer 4omw [Part 1/2]**
>
> We thank the reviewer for the thoughtful and constructive comments. We have uploaded a revised version of our paper with added analyses and kindly invite the reviewer to take a look. Below, we respond to the reviewer’s concerns point by point.
>
> **Q1: Why can the method claim that identity and age conditions are decoupled?**
>
> The “decoupling’’ we refer to is actually a procedural separation of processing stages intended to minimize mutual interference rather than an explicit disentanglement of identity and age features: **(1)** Our decoupling aims to reduce the influence of age traits from reference images and to address the lack of cross-age identity-paired datasets as described in Appendix B. **(2)** This procedural separation is implemented by first learning identity information as a stable representation during training, and applying age control only afterward. The design of our age-guidance residual in Eq. (8): $\nabla_{z_t} L_{age} = \epsilon_\theta(z_t, I_{LQ}, I_{Ref}, c', t) - \epsilon_\theta(z_t, I_{LQ}, I_{Ref}, c, t)$ is also intended to isolate non-age components, ensuring that the identity condition $I_{Ref}$ is not overwritten during inference. **(3)** Our method does not attempt explicit disentanglement of identity and age features, since identity- and age-related cues naturally overlap in facial regions such as the eyes or nose, and the latent space of Diffusion Models intrinsically does not provide clean feature decomposition (Song et al., 2020), making such orthogonal separation inherently difficult.
>
> **Q2: Do identity features remain robust during age-aware gradient guidance when reference images span wide age gaps?**
>
> To assess the robustness of identity features under large age gaps, we conduct an additional analysis presented in Appendix F (“Stability of Identity Conditioning under Age-Aware Gradient Guidance”). Please refer to the updated paper for detailed visualizations and discussion. By visualizing the identity tokens cross-attention maps under reference images of different ages, we observe that the resulting maps exhibit nearly identical spatial activation patterns—consistently focusing on identity-defining facial region—indicating that the learned identity features remains highly stable and is not affected by the age-aware guidance.
>
> **Q3: What is the actual computational overhead for MeInTime during inference? Is the method practical for real-time or high-throughput use cases?**
>
> We report the inference time of all diffusion-based restoration methods under unified 50-step DDIM sampling setup:
> | Methods          | DifFace | Dr2  | RestorerID | Ref-LDM | FaceMe | MeInTime (N=0) | MeInTime (N=5) |
> |------------------|---------|------|------------|---------|--------|----------------|----------------|
> | Time (s)         | 5.96    | 1.23 | 7.74       | 1.79    | 7.65   | 7.26           | 43.37          |
>
> In the same-age setting (N=0), MeInTime takes 7.26s per image, which is comparable to FaceMe (7.65s) and RestorerID (7.74s). In the cross-age setting with our recommended N=5 age-guidance steps, the runtime increases to 43.37s. We think this additional cost is acceptable in practice, as cross-age restoration is generally a low-frequency but high-value scenario: typically arising in historical archive recovery or forensic identity verification, where the images are usually processed individually rather than in bulk. In such settings, achieving accurate restoration is far more critical than real-time latency.
>
> To put this overhead into perspective, it is helpful to consider the complexity of the task we tackle. Our method performs cross-age face restoration, which requires recovering faithful identity while simultaneously handling age-related variations. In this sense, the age editing procedure can be viewed as a sub-task of our approach. For reference, Fading—the most effective age editing pipeline based on DDIM inversion—requires approximately 133.41s per image (Appendix H). In contrast, our unified MeInTime completes the entire restoration in 43.37s, despite addressing a more complex task, and is still more efficient while achieving superior results.
>
> **Q4: What is the effect of the GRF module hyperparameters?**
>
> The GRF module does not introduce any manually tuned hyperparameters. As defined in Eqs. (4–6), GRF learns a gating weight $G^i$ from the concatenated structural features $F^i_{LQ}$ and skip features $F^i_{skip}$ from the encoder of UNet. The influence strength of $F^i_{LQ}$ is fixed to 1.0, following the base model (DiffBIR) design. The fused output $F^i_{out}$ is then concatenated with the decoder features without extra scaling. In other words, GRF is a parameterized block whose behavior is learned end-to-end. Its effect is evaluated in ablation study on Sec.4.3 ("Effectiveness of Gated Residual Fusion"), showing that GRF stabilizes the interaction between structural and identity features and enables more visually natural and identity-preserving restoration.

---

> ### Author Response · Authors · 2025-11-25
> **Response to Reviewer 4omw [Part 2/2]**
>
> **Q5: What is the initialization process for identity projection?**
>
> The identity projection network $P$ is implemented as a lightweight Resampler module (51M parameters) that maps the identity embedding $e \in \mathbb{R}^{512}$ into a set of identity tokens $f \in \mathbb{R}^{16 \times 1024}$ to match the UNet cross-attention feature space. As described in Sec.3.2 ("Face Embedding"), we choose 16 identity tokens to allow finer-grained identity conditioning. The Resampler adopts a Perceiver-style formulation: a small set of learnable latent tokens is iteratively refined through transformer-style attention and feed-forward layers conditioned on the identity embedding, and then projected to the 1024-dim space. All parameters are initialized with PyTorch defaults and trained end-to-end; no manual hyperparameters or special initialization is required.
>
> **Q6: What is the mechanism of the scaling of guidance during inference?**
>
> Our timestep-scaled modulation term $\sqrt{\alpha_t}$ is the standard timestep-dependent coefficient in DDIM/DDPM, which represents the signal component of the latent and decreases as the timestep increases. We notice that model's prediction at very noisy timestep (e.g., $t = T$) is far from a natural face and therefore provides unreliable age semantics. As the timestep decreases, the prediction becomes progressively more structured and begins to capture fine-grained facial semantics. Consequently, $\sqrt{\alpha_t}$ naturally produces the desired modulation behavior: it suppresses guidance when the latent is dominated by noise and gradually strengthens it as age semantics emerge.
>
> The scalar $\lambda$ in Algorithm 1 is a global scaling factor to amplify the guidance gradient. Following common practice in SDS-style editing method (Nam et al., 2024), we set $\lambda = 1000$ after performing a small validation sweep and keep this value fixed for all experiments.
>
> **Q7: Could Algorithm 1 be further clarified?**
>
> We thank the reviewer for the insightful suggestion. We have revised Algorithm 1 to clarify the procedure, and kindly invite the reviewer to check the updated paper.
>
> **Q8: Does the current submission hinder reproducibility?**
>
> We appreciate the reviewer’s concern regarding reproducibility. We plan to release the code and pretrained models upon acceptance, which will fully support verification and further research.
>
> **Reference:**
>
> - Song *et al.*, 2020. *Score-based generative modeling through stochastic differential equations.*
>
> - Nam *et al.*, 2024. *Contrastive denoising score for text-guided latent diffusion image editing.*

---

> ### Comment · Reviewer_4omw · 2025-11-28
>
> Thanks for your detailed responses, which somehow addressed my concerns. I have decided to raise my score.

---

> > ### Author Response · Authors · 2025-11-29
> > **Response to Reviewer 4omw**
> >
> > We sincerely appreciate your willingness to raise the score and your constructive feedback throughout the review process. We are very glad that our responses and additional analyses have helped resolve your earlier concerns.

---

### Official Review · Reviewer_JapA · 2025-11-01

**Soundness:** 3
**Presentation:** 3
**Contribution:** 2
**Rating:** 6
**Confidence:** 3

**Summary:**

This paper introduces MeInTime, a novel diffusion-based framework that significantly advances reference-based face restoration by effectively tackling the challenging cross-age scenario. The key innovation lies in its decoupled modeling of identity and age conditions: it injects identity features via a dedicated attention mechanism during training, and at inference, employs a training-free Age-Aware Gradient Guidance to steer the generation towards the target age. Extensive experiments confirm that MeInTime outperforms existing methods, achieving superior identity fidelity and age consistency simultaneously.

**Strengths:**

+Pioneering Cross-Age Reference-Based Framework: This work introduces the first reference-based face restoration framework specifically designed for cross-age scenarios, effectively extending the capability of existing methods from same-age to cross-age restoration by incorporating target age prompts.
+ Novel Disentangled Training-Inference Strategy: The proposed method employs a decoupled approach that separately handles identity preservation during training through dedicated attention mechanisms, and age consistency during inference via training-free Age-Aware Gradient Guidance, effectively resolving identity-age conflicts.
+ Gated Residual Fusion modules dynamically integrate structural features from degraded inputs with identity representations in a content-aware manner.
+ Plug-and-play Age-Aware Gradient Guidance steers generation toward target age semantics without retraining.
+ Comprehensive experimental validations show superior performance in visual quality, identity preservation, and age consistency compared to existing approaches.

**Weaknesses:**

- According to Table 1, the performances of the proposed method are not always the best. The authors should explain the reasons in detail.
- The authors do not compare the speed and the number of parameters of the proposed method, compared to existing methods.
- The authors do not present the failure cases of the proposed method. I think it is better to analyze the limitations.
- The fonts in the Figures are too small.
- The effectiveness of the Age-Aware Gradient Guidance is not verified in the ablation studies.

**Questions:**

Please see the weaknesses.

---

> ### Author Response · Authors · 2025-11-27
> **Response to Reviewer JapA [Part 1/2]**
>
> We sincerely appreciate the reviewer’s insightful comments and constructive suggestions. We have uploaded a revised version of our paper with added analyses and kindly invite the reviewer to check. Our responses to all specific concerns are presented below.
>
> **Q1: Why the performances of the proposed method are not always the best as shown in Table 1?**
>
> We thank the reviewer for this insightful question. The quantitative experimental analysis in Table 1 focuses on three aspects of performance, which are our restoration goals: image quality, identity similarity, and age consistency. MeInTime achieves the best results on identity similarity (IDS) and age consistency (AGE) in both same-age and cross-age settings.
>
> Regarding image quality, in the **same-age setting** MeInTime shows the strongest overall performance: it achieves the best PSNR and LPIPS, the second-best SSIM and MUSIQ, and a highly competitive FID. In the **cross-age setting**, MeInTime attains the best image quality among all **reference-based** methods. When compared with the best **reference-free** method CodeFormer, which optimizes purely for image quality, our approach still attains comparable perceptual fidelity (LPIPS, FID) and visual naturalness (MUSIQ), despite simultaneously optimizing identity similarity and age consistency.
>
> However, on the cross-age set we observe a slight drop in PSNR/SSIM, which are pixel-level structural metrics. We analyze the performance drop arises from our age-aware gradient guidance: the iterative global update used to correct the target-age semantics inevitably spreads gradient over the whole image and may introduce mild deviations in image structures, as shown in Eq.(10): $z_{t-1}' = z_{t-1} - \sqrt{\alpha_t} \cdot \nabla_{z_t} L_{age}$.  We have added a brief discussion of this limitation in Appendix I of the revised paper, and we kindly invite the reviewer to refer to the visual analysis. We thank the reviewer again for this helpful observation, and mitigating these structural side effects will be an important direction for our future work.
>
> **Q2: What is the speed and the number of parameters of the proposed method, compared to existing methods?**
>
> We report the inference time of all diffusion-based restoration methods under unified 50-step DDIM sampling setup:
>
> | Methods | DifFace | Dr2  | RestorerID | Ref-LDM | FaceMe | MeInTime(N=0) | MeInTime(N=5) |
> | ------- | ------- | ---- | ---------- | ------- | ------ | ------------- | ------------- |
> | Time(s) | 5.96    | 1.23 | 7.74       | 1.79    | 7.65   | 7.26          | 43.37         |
>
> In the same-age setting (N=0), MeInTime takes 7.26s per image, which is comparable to FaceMe (7.65s) and RestorerID (7.74s). In the cross-age setting with our recommended N=5 age-guidance steps, the runtime increases to 43.37s. We think this additional cost is acceptable in practice, as cross-age restoration is generally a low-frequency but high-value scenario: typically arising in historical archive recovery or forensic identity verification, where the images are usually processed individually rather than in bulk. In such settings, achieving accurate restoration is far more critical than real-time latency.
>
> To put this overhead into perspective, it is helpful to consider the complexity of the task we tackle. Our method performs cross-age face restoration, which requires recovering faithful identity while simultaneously handling age-related variations. In this sense, the age editing procedure can be viewed as a sub-task of our approach. For reference, Fading—the most effective age editing pipeline based on DDIM inversion—requires approximately 133.41s per image (details provided in Appendix H). By comparison, our unified MeInTime framework completes the entire restoration in 43.37s, despite addressing a significantly more complex task, and is still substantially more efficient while achieving superior results.
>
> For model parameters, we are unfortunately unable to obtain the complete parameter counts of some baselines. For transparency, we provide the full parameter breakdown of MeInTime:
>
> - GRF module (Sec. 3.2 "Gated Residual Fusion module"): 25M
> - Additional Key/Value projection layers in UNet (Sec. 3.2 "Feature Injection"): 34M
> - Projection Network $P$ (Sec. 3.2 "Face Embedding"): 51M
>
> yielding a total of 110M parameters, which is approximately 10–12% of the entire SD model.

---

> > ### Author Response · Authors · 2025-11-27
> > **Response to Reviewer JapA [Part 2/2]**
> >
> > **Q3: Are there failure cases of the proposed method? What are the limitations?**
> >
> > We thank the reviewer for this valuable question. In the same-age restoration setting, MeInTime rarely exhibits noticeable failures. In the cross-age setting, however, a few imperfect cases may appear, which mainly relate to the usage of the Age-Aware Gradient Guidance. This is consistent with our analysis in **Q1**. Specifically, the age-control mechanism relies on textual prompts to guide age semantics; however, textual age descriptions do not always perfectly match human visual perception of age. When conditioning on very high age targets, the iterative global optimization may occasionally introduce over-sharpened or exaggerated facial structure textures. We provide several representative examples of such cases in Appendix I of the revised paper, and we kindly invite the reviewer to refer to these qualitative results. Addressing these structural side effects and improving age-aware guidance robustness will be an important direction of our future work.
> >
> > **Q4: The fonts in the Figures are too small.**
> >
> > We thank the reviewer for pointing out this issue. Due to the space constraints in the initial submission, some figure texts were presented at a smaller scale. In the revised version, we have made the corresponding adjustments to ensure a better reading experience.
> >
> > **Q5: The effectiveness of the Age-Aware Gradient Guidance is not verified in the ablation studies.**
> >
> > We thank the reviewer for raising this point. Our ablation studies indeed include a detailed analysis of the effectiveness of the proposed Age-Aware Gradient Guidance (AAGG). To clarify the potential misunderstanding, we summarize the relevant experiments below. **(1) Sec. 4.3 “Inference Strategy”** directly examines the effect of applying AAGG during inference. In Fig.10 and Tab.3, we refer to not applying AAGG as Age Prompt, which performs a single forward pass using only the textual age description (e.g., “photo of a 24-year-old person”), while Age Guidance corresponds to enabling AAGG for iterative age-controlled generation as shown in Algorithm 1. Both quantitative and qualitative experimental results show that Age Aware Gradient Guidance achieves significantly better age alignment. **(2) Sec. 4.3 “Robustness on Varying Age Gaps”** further examines the robustness of AAGG when the ages of the reference and degraded inputs differ by various intervals. Tab.2 shows all metrics remain stable across intervals, indicating that AAGG is resilient to large age gaps. **(3) Sec. 4.3 “Timestep-Scaled Modulation Term”** evaluates the internal design choices of AAGG by comparing fixed scales (0, 0.5, 1.0) against our timestep-adaptive modulation $\sqrt{\alpha_t}$ as defined in Eq.(10):  $z_{t-1}' = z_{t-1} - \sqrt{\alpha_t} \cdot \nabla_{z_t} L_{age}$.  As shown in Fig.12, fixed scales either under-correct or over-correct the age semantics, while the adaptive strategy achieves a better balance between semantic precision and visual fidelity.
> >
> > Taken together, these three ablations comprehensively verify the necessity and effectiveness of the proposed AAGG from multiple perspectives—whether to apply it, how stable it is across age gaps, and how to modulate its strength. We hope this clarifies the experimental coverage and kindly invite the reviewer to refer to Sec. 4.3 for full details.

---

### Author Response · Authors · 2025-11-28
**General Response [Part 1/2]**

We sincerely thank all reviewers for their thoughtful and constructive feedback, which has greatly helped us strengthen the paper. Due to the unexpected situation at ICLR 2026, reviewers were unable to continue the rebuttal discussion. Prior to this interruption, two reviewers ***(4mow, 4pTD)*** had already responded to our rebuttal. Both confirmed that our clarifications satisfactorily addressed their concerns and expressed their willingness to raise their scores.

Below, we provide a consolidated summary of our key contributions, address the common concerns raised across the reviews, and highlight the major revisions incorporated in the updated manuscript, to facilitate a quick and clear assessment by the new area chair.

## **Primary Contributions:**

- We present the first reference-based face restoration framework specifically designed for cross-age scenarios, extending the scope of identity-preserving restoration beyond conventional same-age settings.
- We design a disentangled strategy that handles identity preservation during training and enforces age consistency during inference, effectively mitigating identity-age conflicts and alleviating cross-age data scarcity.
- The proposed Gated Residual Fusion modules dynamically integrate structural features from the degraded input with identity representations in a content-aware manner, enabling robust identity incorporation.
- Our plug-and-play Age-Aware Gradient Guidance strategy steers the generative process toward the desired age manifold through a prompt-driven optimization procedure without requiring additional training.
- Experiments on standard same-age benchmark show that our method outperforms existing approaches in both visual quality and identity preservation. Experiments on cross-age and real-world sets further validate our superior performance in terms of visual quality, identity preservation, and age consistency.

## **Common Concerns and Revisions:**

**Evaluation on the real-world dataset** ***(Reviewer ojSk)*:**

We have added a comprehensive real-world evaluation in **Appendix G ("Evaluation on the Real-world dataset")**, including dataset construction, evaluation protocols, and both quantitative and qualitative comparisons. The experiment results show that MeInTime achieves strong visual quality, identity preservation, and age consistency, ranking second in MUSIQ and first in FID, IDS, and AGE, with the AGE score improved by 17.4% over the second-best method, demonstrating its robustness and effectiveness under real-world conditions.

**Subjective evaluation** ***(Reviewers ojSk, 4pTD)*:**

We have added a dedicated user study in **Sec. 4.2 ("User Study")** of the revised paper and provided detailed questionnaire design in **Appendix K ("Complete User Study Content")**. These results indicate that MeInTime maintains high visual quality while delivering the most faithful identity preservation and the most reliable age consistency in human-centric evaluation.

**Inference time** ***(Reviewers JapA, 4omw, ojSk)*:**

We report the inference time of all diffusion-based restoration methods under a unified 50-step DDIM sampling setup in **Appendix I ("Limitations")**. MeInTime achieves comparable speed to reference-based methods in the same-age setting. In cross-age setting where the optimization procedure is performed, MeInTime remains more efficient than two-stage pipelines (which first restores the face and then performs age editing), achieving 43.37 s vs. 133.41 s while still producing superior results.

**Limitations** ***(Reviewers JapA, ojSk)*:**

We summarize the limitations and several undesired examples in **Appendix I ("Limitations")** of the revised paper, along with detailed analysis and visual results. Addressing these effects will be an important direction for our future work.

**Ablation study on the robustness of identity embedding** ***(Reviewer 4omw)*:**

We conduct an additional analysis in **Appendix F (“Stability of Identity Conditioning under Age-Aware Gradient Guidance”)** to assess the robustness of identity features under large age gaps. By visualizing the cross-attention maps of identity tokens under reference images of different ages, we observe that the resulting maps exhibit nearly identical spatial activation patterns, indicating that the learned identity features remain highly stable and are not affected by the age-aware guidance.

---

> ### Author Response · Authors · 2025-11-30
> **General Response [Part 2/2]**
>
> **Evaluation on Two-Stage Restoration–Editing Pipelines** ***(Reviewer 4pTD)*:**
>
> We further conduct an experiment in **Appendix H ("Evaluation on Two-Stage Restoration–Editing Pipelines")**, where we evaluate a two-stage workflow that first restores the face and then applies age editing to the restored result as a direct comparison to MeInTime. Appendix H provides details on the experimental setup, data selection, evaluation metrics, and both quantitative and qualitative comparisons. The results show that MeInTime surpasses all post-editing baselines by a large margin across visual quality, identity similarity, and age alignment.
>
> **Reproducibility Considerations and Code Release** ***(Reviewer 4omw)*:**
>
> We have added an anonymous link at the end of the **Abstract** that provides the full inference pipeline of MeInTime. Upon acceptance, we will release the complete training code and pretrained models to ensure full reproducibility and facilitate future research.
>
> We sincerely appreciate the time and thoughtful effort you have invested in reviewing our work. We hope that the revisions and detailed responses provided below satisfactorily address your concerns.

---

### Meta-Review · Area_Chair_CbA1 · 2026-01-03

**Summary:**

The paper proposes a diffusion-based reference-guided face restoration method aimed at addressing cross-age gap between reference and input images. The submission received a borderline average rating of 5.0 (4, 4, 6, 6). Reviewers raised concerns about inferior performance compared to prior methods (JapA, ojSk) and several missing components in the evaluation and discussion, including inference-time reporting (JapA, 4omw, ojSk), a limitations discussion (JapA, ojSk), evaluations on real-world data (ojSk), subjective user studies (ojSk, 4pTD), and comparison with two-stage restoration–editing pipelines (4pTD). Additional concerns include the effectiveness of the ablation studies (JapA, 4omw) and the practical importance of the cross-age setting (ojSk).

After carefully reading the paper, the reviews, and the rebuttal, I agree that the authors addressed several of the raised concerns, and the problem setting itself is interesting and meaningful. However, some key concerns regarding performance and efficiency remain in the current version. In particular, the visual quality of the proposed method is not yet fully convincing in both quantitative and qualitative evaluations. Moreover, the inference time in the cross-age setting (43.37s per image) raises concerns about practical applicability. Considering these remaining concerns regarding performance robustness and efficiency, the AC tends to recommend Reject at this stage.

**Reviewer Concerns:**

The authors provided additional experiments, evaluations, and comparisons in the rebuttal and revised supplementary material.  Most of the concerns listed above were adequately addressed. However, key concerns regarding performance and efficiency remain are still outstanding in this version, as detailed below:

1. Visual quality: The visual quality of the proposed method is not yet fully convincing in both quantitative and qualitative evaluations. As reported in Table 1, the method underperforms strong non-reference-based prior approaches such as CodeFormer across multiple metrics, and the user study further indicates that its visual results are not consistently preferred. Qualitatively, the provided results often exhibit artifacts and unrealistic teeth around the mouth region (e.g., Figs. 6, 10, 12, 16, and 18). As also discussed by the authors, this quality drop may stem from the age-aware gradient guidance, suggesting intrinsic limitations of the current method.

2. Efficiency: The inference time of the proposed method for cross-age setting is approximately 6x slower than other diffusion-based approaches. In addition, the efficiency comparison in the rebuttal does not include the strong non-diffusion baselines, which are likely to be more efficient. This raises concerns about the practical applicability of the method in real-world scenarios.

**Reviewer Scores:**

Reviewers 4omw (score: 4) and 4pTD (score: 6) has responded during the rebuttal phase and indicated a willingness to raise their scores. However, Reviewer 4omw’s response came after the system bug and was therefore only taken as a reference in the final decision.

Reviewer JapA (score: 6) and Reviewer ojSk (score: 4) had some of their concerns partially addressed in the rebuttal. However, given the remaining concerns, they would likely maintain their initial ratings.

---

### Decision · Program_Chairs · 2026-01-26

Reject